# The β-hairpin of 40S exit channel protein Rps5/uS7 promotes efficient and accurate translation initiation in vivo

Jyothsna Visweswaraiah, Yvette Pittman, Thomas E Dever*, Alan G Hinnebusch*

Laboratory of Gene Regulation and Development, National Institute of Child Health and Human Development, National Institutes of Health, Bethesda, United States

**Abstract** The eukaryotic 43S pre-initiation complex bearing tRNA$_i^{Met}$ scans the mRNA leader for an AUG start codon in favorable context. Structural analyses revealed that the β-hairpin of 40S protein Rps5/uS7 protrudes into the 40S mRNA exit-channel, contacting the eIF2•GTP•Met-tRNA$_i$ ternary complex (TC) and mRNA context nucleotides; but its importance in AUG selection was unknown. We identified substitutions in β-strand-1 and C-terminal residues of yeast Rps5 that reduced bulk initiation, conferred 'leaky-scanning' of AUGs; and lowered initiation fidelity by exacerbating the effect of poor context of the eIF1 AUG codon to reduce eIF1 abundance. Consistently, the β-strand-1 substitution greatly destabilized the '$P_{IN}$' conformation of TC binding to reconstituted 43S·mRNA complexes in vitro. Other substitutions in β-hairpin loop residues increased initiation fidelity and destabilized $P_{IN}$ at UUG, but not AUG start codons. We conclude that the Rps5 β-hairpin is as crucial as soluble initiation factors for efficient and accurate start codon recognition.

*For correspondence: Thomas. Dever@nih.gov (TED); ahinnebusch@nih.gov (AGH)

Competing interests: The authors declare that no competing interests exist.

## Introduction

Accurate identification of the translation initiation codon is critical to ensure synthesis of the correct cellular proteins. In eukaryotic cells this process generally occurs by a scanning mechanism, wherein the small (40S) ribosomal subunit first recruits initiator tRNA (Met-tRNA$_i$) in a ternary complex (TC) with eIF2-GTP in a reaction stimulated by eIFs 1, 1A, and 3. The resulting 43S pre-initiation complex (PIC) attaches to the mRNA 5′ end and scans the 5′ UTR for an AUG, using complementarity with the anticodon of Met-tRNA$_i$ to identify the start codon and assemble a 48S PIC. Nucleotides immediately surrounding the AUG, particularly the −3 and +4 positions (referred to below as context nucleotides), also influence start codon selection. During scanning, the GTP bound to eIF2 in the TC is hydrolyzed in the 43S PIC in a manner dependent on the GTPase activating protein eIF5, but $P_i$ release is blocked by eIF1, which also impedes stable binding of Met-tRNA$_i$ in the P site. Start codon recognition triggers dissociation of eIF1 from the 40S subunit, which allows interaction between eIF5 and the C-terminal tail (CTT) of eIF1A, $P_i$ release from eIF2-GDP·$P_i$, and more stable TC binding in the P site (*Figure 1*). Subsequent dissociation of eIF2-GDP and other eIFs from the 48S PIC enables eIF5B-catalyzed subunit joining and formation of an 80S initiation complex with Met-tRNA$_i$ base-paired to AUG in the P site (reviewed in *Hinnebusch, 2014*).

eIF1 plays a dual role in the scanning mechanism. It promotes an open, scanning-conducive conformation of the PIC (*Pestova and Kolupaeva, 2002*) to which TC rapidly loads, bound in a state capable of inspecting successive triplets entering the P site (dubbed $P_{OUT}$) (*Passmore et al., 2007*; *Saini et al., 2010*); and it also blocks recognition of near-cognate start codons (e.g., UUG) (*Yoon and Donahue, 1992*) and AUG codons in poor sequence context (*Pestova and Kolupaeva, 2002*). Hence, eIF1 must dissociate from the 40S subunit (*Maag et al., 2005*; *Cheung et al., 2007*) to allow $P_i$ release (*Algire et al., 2005*) and rearrangement to a scanning-incompatible state with Met-tRNA$_i$ base paired

**eLife digest** To make a protein, the DNA sequence of a gene is first copied to make an mRNA molecule, which is then translated into a protein by a molecular machine called the ribosome. The first step of translation is known as initiation. Several proteins referred to as initiation factors can bind to the small subunit of the ribosome, which itself is composed of an RNA molecule and numerous proteins, and form a pre-initiation complex (or PIC for short). This complex contains a molecule called initiator tRNA that is specialized for initiation. The PIC then attaches to the mRNA and starts scanning it, searching the sequence for an AUG triplet to serve as the start codon. The sequence immediately surrounding an AUG triplet, known as the context, influences the likelihood of its selection as the start site. When the start codon is recognized by the initiator tRNA in the PIC, the complete ribosome assembles and begins to build the protein. Choosing the correct start codon is crucial to ensure that the correct protein is made from every mRNA in the cell.

The PIC can adopt an 'open' state, which makes it easier to scan the mRNA for the correct start codon and ignore triplets similar in sequence to AUG (like UUG) or AUG triplets in a poor context. Once the correct AUG start codon has been recognized, the PIC changes to a 'closed' state, ceases to scan, and assembles the complete ribosome.

One of the proteins that make up the small ribosomal subunit (called Rps5 in budding yeast) is located near the channel where the mRNA molecule exits the PIC during scanning, and is also thought to be involved in translation initiation. However, the role of Rps5 in recognizing the start codon is poorly understood.

Visweswaraiah et al. have now studied Rps5—in particular, a region of this protein that adopts a hairpin structure that dips into the exit channel—using genetic and biochemical methods. In mutant yeast cells in which the hairpin structure was mutated, translation initiation was diminished at suboptimal start codons—including a UUG start codon and an AUG codon in poor context—thus making translation initiation more accurate.

Visweswaraiah et al. then performed experiments on PICs built from purified components to determine how the Rps5 mutations might affect the assembly and stability of the PIC. The results revealed that mutating the upper region of the Rps5 hairpin destabilized the closed state of the PIC when either an AUG or UUG start codon was present in the mRNA. However, other mutations of the hairpin structure destabilized the closed state of the PIC only at a UUG start codon. In both cases, the mutations made the PIC more likely to remain in the open conformation and continue scanning at incorrect or suboptimal start codons, making it more likely that translation begins at the correct AUG start codon.

These results indicate that the Rps5 hairpin is crucial for both efficiently and accurately recognizing the start codon to begin translation. This suggests that ribosomal proteins not only contribute to ribosome structure but can actively participate with other initiation factors in choosing the correct start sites for protein synthesis on all mRNAs in the cell.

with AUG and more tightly bound in the $P_{IN}$ conformation (*Passmore et al., 2007*; *Saini et al., 2010*). Consistent with this, structural analyses of different PICs reveal that eIF1 and eIF1A promote rotation of the 40S head relative to the body (*Lomakin and Steitz, 2013*) (*Hussain et al., 2014*), which is likely instrumental in TC binding in the $P_{OUT}$ conformation, but that eIF1 physically obstructs Met-tRNA$_i$ binding in the $P_{IN}$ state (*Rabl et al., 2011*; *Lomakin and Steitz, 2013*). Accordingly, eIF1 is deformed and displaced from its 40S location in the open complex during the $P_{OUT}$ to $P_{IN}$ transition (*Hussain et al., 2014*). Consequently, mutations that weaken eIF1 binding to the 40S subunit reduce the rate of TC loading, while elevating initiation at near-cognate codons or AUGs in poor context, by destabilizing the open/$P_{OUT}$ conformation and favoring rearrangement to the closed/$P_{IN}$ state during scanning (*Martin-Marcos et al., 2011*, *2013*). Moreover, decreasing wild-type (WT) eIF1 abundance reduces initiation accuracy, whereas overexpressing eIF1 suppresses initiation at near cognates or AUGs in poor context (*Valasek et al., 2004*; *Alone et al., 2008*; *Ivanov et al., 2010*; *Saini et al., 2010*; *Martin-Marcos et al., 2011*). This tight link between eIF1 abundance and initiation accuracy is exploited to autoregulate eIF1 expression, as the AUG start codon of the eIF1 gene (*SUI1* in yeast) occurs in poor context—a feature conserved throughout eukaryotic evolution—and the frequency

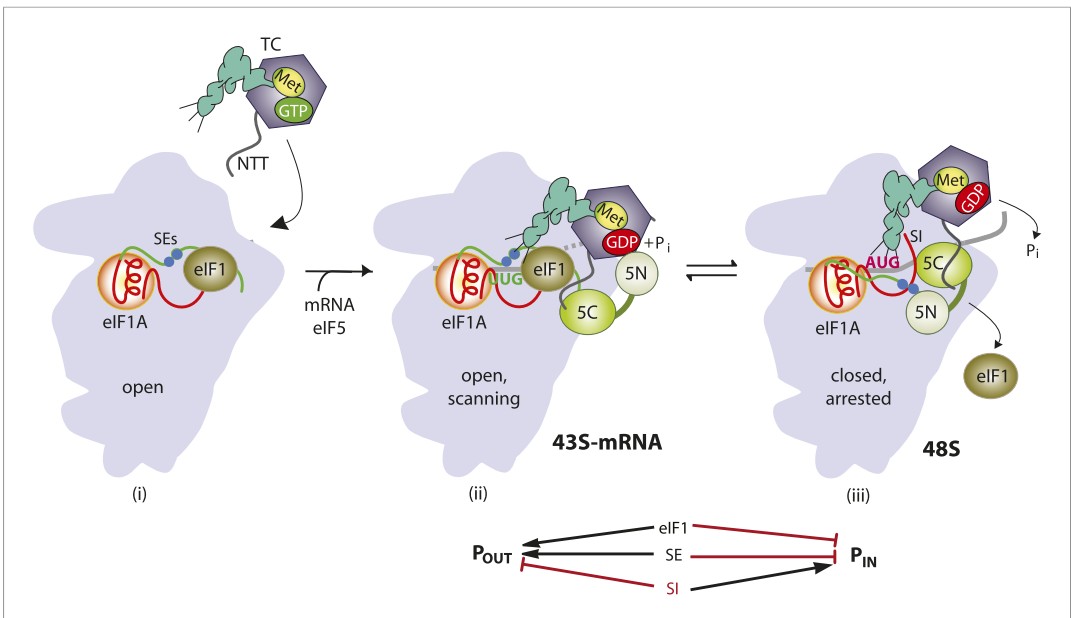

**Figure 1**. Model describing conformational rearrangements of the PIC during scanning and start codon recognition. Assembly of the PIC, scanning and start codon selection in WT cells. (i) eIF1 and the scanning enhancer (SEs) elements in the CTT of eIF1A stabilize an open conformation of the 40S subunit to which the TC loads rapidly. (ii) The 43S PIC in the open conformation scans the mRNA for the start codon with Met-tRNA$_i$ bound in the $P_{OUT}$ state. The GAP domain in the N-terminal domain of eIF5 (5N) stimulates GTP hydrolysis by the TC to produce GDP•Pi, but release of $P_i$ is blocked. The unstructured NTT of eIF2β interacts with eIF1 to stabilize eIF1•40S association and the open conformation. (iii) On AUG recognition, Met-tRNA$_i$ moves from the $P_{OUT}$ to $P_{IN}$ state, clashing with eIF1 and the CTT of eIF1A. Movement of eIF1 and the eIF1A CTT away from the P site disrupts eIF1's interaction with eIF2β-NTT, and the latter interacts with the eIF5-CTD. eIF1 dissociates from the 40S subunit, and the eIF5-NTD disengages from eIF2 and interacts with the eIF1A CTT instead, dependent on the SE elements, thereby facilitating $P_i$ release from eIF2. The eIF5-CTD moves into the position on the 40S subunit previously occupied by eIF1 and blocks reassociation of eIF1. (Below) Arrows summarize that eIF1 and the eIF1A SE elements promote $P_{OUT}$ and block transition to the $P_{IN}$ state, whereas the scanning inhibitor (SI) element in the NTT of eIF1A stabilizes the $P_{IN}$ state. (Adapted from *Hinnebusch and Lorsch, 2012*; *Nanda et al., 2013*).

of recognizing its own start codon is inversely related to eIF1 abundance (*Ivanov et al., 2010*; *Martin-Marcos et al., 2011*).

The stability of the codon-anticodon duplex is an important determinant of initiation accuracy, as the rate of the $P_{OUT}$ to $P_{IN}$ transition and stability of the $P_{IN}$ state are both favored by AUG vs non-AUG start codons (*Kolitz et al., 2009*). It is possible that favorable context also contributes to the stability of $P_{IN}$ (*Pisarev et al., 2006*; *Martin-Marcos et al., 2011*), but the stimulatory effect of optimum context on initiation rate is not understood at the molecular level. There is evidence that the context nucleotides are recognized by the α-subunit of eIF2, as replacement of heterotrimeric eIF2 with the eIF2βγ heterodimer reduced the efficiency of AUG recognition and diminished the stimulatory effect of optimum context on 48S PIC assembly in a reconstituted mammalian system (*Pisarev et al., 2006*). Moreover, crosslinking experiments (*Pisarev et al., 2006*; *Sharifulin et al., 2013*) and structural analyses of a mammalian 43S PIC (*Hashem et al., 2013*) and a partial yeast (py48S) PIC (*Hussain et al., 2014*) indicate that the N-terminal domain (D1) of eIF2α is in proximity to the −3 nucleotide of the mRNA in the exit channel of the 40S subunit. These and other studies (*Lomakin and Steitz, 2013*) revealed that the conserved β-hairpin of the 40S protein uS7 (Rps5 in yeast) lies in the vicinity of eIF2α-D1 and the −3 nucleotide of mRNA in reconstituted 43S/48S PICs (*Figure 2A,B*); however, functional evidence that eIF2α-D1 and the Rps5 β-hairpin have important roles in start codon recognition in vivo is lacking.

In this report, we establish that the β-hairpin of Rps5 is critically required for both efficient and accurate translation initiation in vivo. Substituting Glu-144 (E144) in β-strand 1 of the hairpin, or the proximal C-terminal residue R225 (*Figure 2B*), confers a marked reduction in the efficiency of AUG

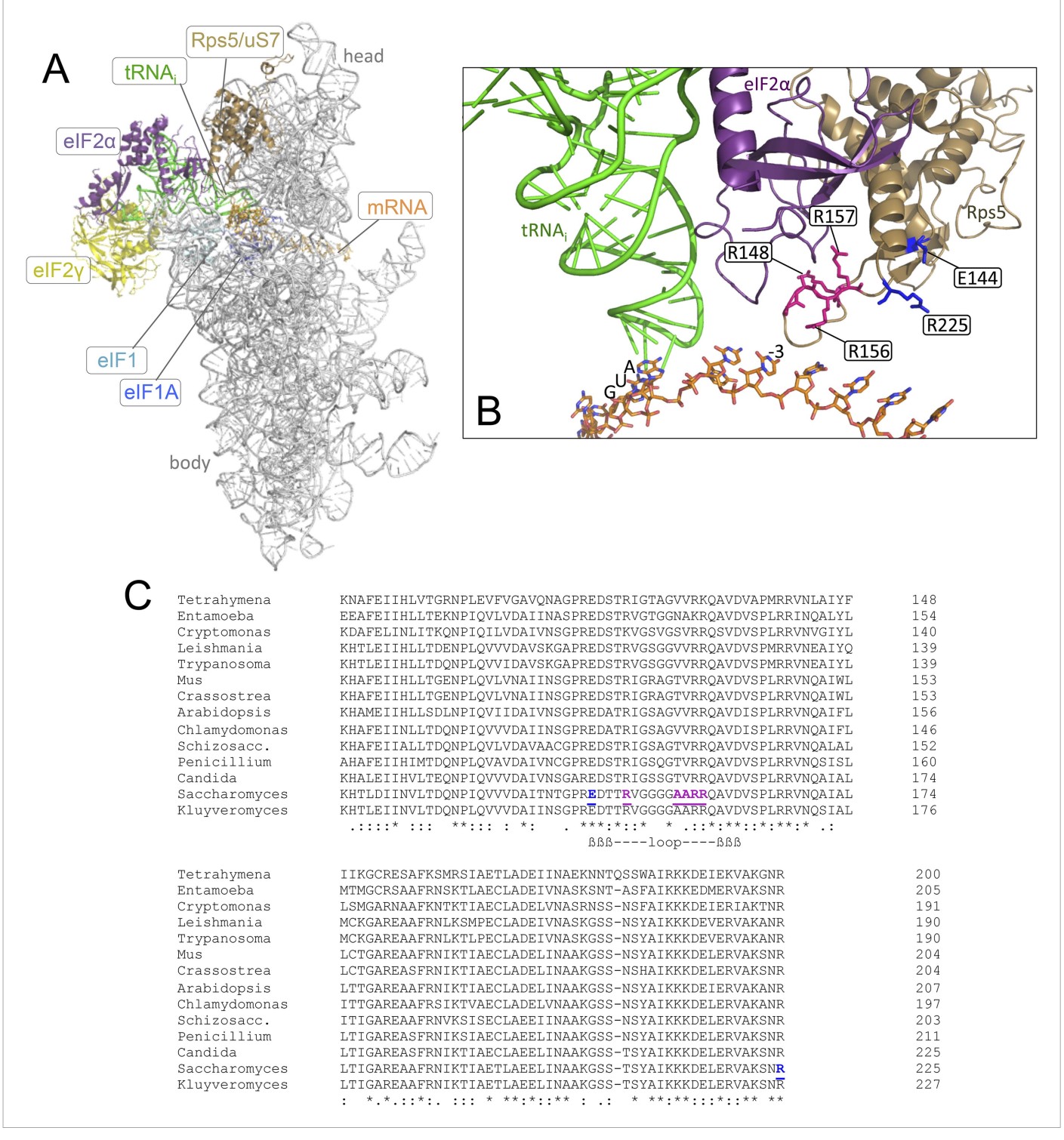

**Figure 2**. Location in the py48S PIC, and sequence conservation, of the Rps5 β-hairpin loop. (**A**, **B**) Depiction of the partial yeast 48S PIC (PDB 3J81) showing Rps5 (gold), mRNA (orange), Met-tRNAᵢ (green), eIF2α (purple), eIF2γ (yellow), eIF1 (cyan) and eIF1A (blue). For clarity other ribosomal proteins, eIF2β and putative eIF5 densities are not shown. Residues implicated here in AUG recognition are shown with stick side-chains and highlighted (as in panel **C**) in blue or pink. (**C**) Alignment of Rps5 sequences from diverse eukaryotes using the Clustal Omega algorithm (http://www.ebi.ac.uk/Tools/msa/clustalo/). The β- hairpin loop is annotated below the alignment and residues implicated in this study in AUG recognition are highlighted in blue or pink and underlined.

recognition. This defect exacerbates the effect of poor context at the *SUI1* mRNA start codon to reduce eIF1 cellular abundance and thereby increase recognition of near-cognate UUG codons as a secondary effect. Analysis in the yeast reconstituted system reveals striking destabilization of the $P_{IN}$ state formed at AUG codons by E144R mutant 40S subunits. Substitutions in the loop portion of the Rps5 β-hairpin also destabilize the $P_{IN}$ state, but produce this effect selectively for near-cognate UUG start codons, and thereby dampen UUG initiation in vivo. These findings indicate that the Rps5 β-hairpin functions on a par with soluble initiation factors, such as eIF1, eIF1A and eIF2, to insure efficient and accurate start codon recognition in eukaryotic cells.

## Results

### Substitutions *E144R* and *R225K* impair translation initiation and start codon selection in vivo

To examine the role of the Rps5 β-hairpin in start codon recognition, we introduced single substitutions into 3 residues of β-strand 1 ($_{144}$EDT$_{146}$) and 8 of the 10 residues in the hairpin loop ($_{147}$TR$_{148}$ and $_{151}$GGGARRQ$_{158}$) (*Supplementary file 1*). Residues in the β-strands, and the loop residues proximal to the β-strands, are among the most highly conserved in evolution (*Figure 2C*). We also substituted the last four residues of Rps5 ($_{222}$KSNR$_{225}$) in view of their strong conservation and proximity to the β-hairpin, and because invariant Glu144 in β-strand 1 (E144) appears to form a salt-bridge with C-terminal residue R225 in the yeast 80S ribosome (*Ben-Shem et al., 2011*) (*Figure 2B*). Residues were generally substituted with Ala to shorten the side-chain, or with basic or acidic residues to introduce or alter side-chain charge (*Supplementary file 1*). The mutations were generated in an *RPS5* allele under its own promoter on a low-copy plasmid and examined in a yeast strain with WT chromosomal *RPS5* under a galactose-inducible promoter ($P_{GAL1}$). Mutant phenotypes were scored following a switch from galactose to glucose, where $P_{GAL1}$-*RPS5* expression is repressed. Despite strong sequence conservation of many β-hairpin residues (*Figure 2C*), only the G151S substitution was lethal and prevented growth on glucose medium; however, several substitutions conferred a slow-growth (Slg⁻) phenotype, including *E144R* and *R225K* (*Figure 3A*, glucose; *Supplementary file 1*).

To identify effects on fidelity of start codon selection, the mutant strains were assayed for expression of otherwise identical *HIS4-lacZ* reporters containing an AUG or UUG start codon. Substantial (>twofold) increases in expression of the UUG relative to AUG reporter (UUG:AUG ratio) were observed only for *E144R* and three different mutations substituting Arg225. Mutations *E144R* and *R225K* elevated the UUG:AUG ratio by 5.9- and 3.6-fold, respectively (*Figure 3B* and *Supplementary file 1*). To test the importance of the Glu144/Arg225 salt-bridge, we constructed the *E144R/R225E* double mutant in which the salt-bridge should be reinstated. This strain displayed a Slg⁻ phenotype and increased UUG:AUG ratio similar in magnitude to that of the *R225K* mutant but less severe than seen for the *E144R* strain (*Figure 3A,B*). The fact that combining these mutations did not produce more severe phenotypes than those conferred by *E144R* alone is consistent with the possibility that reinstating the salt-bridge mitigates the effects of the *E144R* single mutation, with the stipulation that the WT identity of E144 or R225 is needed for robust Rps5 function. Thus, although the salt-bridge is reinstated, the substitution of one or both residues still impairs growth and initiation fidelity in the double mutant. Additional experiments are needed to establish the importance of the salt bridge for Rps5 function.

Mutations in various initiation factors are known that elevate the UUG:AUG ratio and restore translation of mutant *his4-301* mRNA, which lacks the AUG start codon, by enabling initiation at the third, UUG codon, thereby suppressing histidine auxotrophy and conferring a Sui⁻ (Suppressor of initiation codon mutation) phenotype. However, neither *E144R* nor *R225K* suppress the His⁻ phenotype of *his4-301* to confer the Sui⁻ phenotype. Based on previous observations of Sui⁻ mutants, it is possible that the Rps5 substitutions do not elevate the UUG:AUG ratio sufficiently to produce enough *his4-301* product for adequate histidine biosynthesis (*Dorris et al., 1995*; *Martin-Marcos et al., 2013*). For example, the eIF1 mutations *sui1-K37A* and *sui1-R33A* increase the UUG:AUG ratio by 4.8- and 7.7-fold, but only the latter suppresses the His⁻ phenotype of *his4-301* (*Martin-Marcos et al., 2013*). Alternatively, the Rps5 mutations could interfere with an unknown aspect of histidine biosynthesis or utilization (*Nanda et al., 2009*).

Consistent with their Slg⁻ phenotypes, *E144R* and *R225K* conferred significant reductions in the polysome:monosome (P/M) ratio (p-value <0.0005) (*Figure 3C*), indicating a reduced rate of bulk translation initiation relative to elongation, with the greater reduction conferred by the mutation

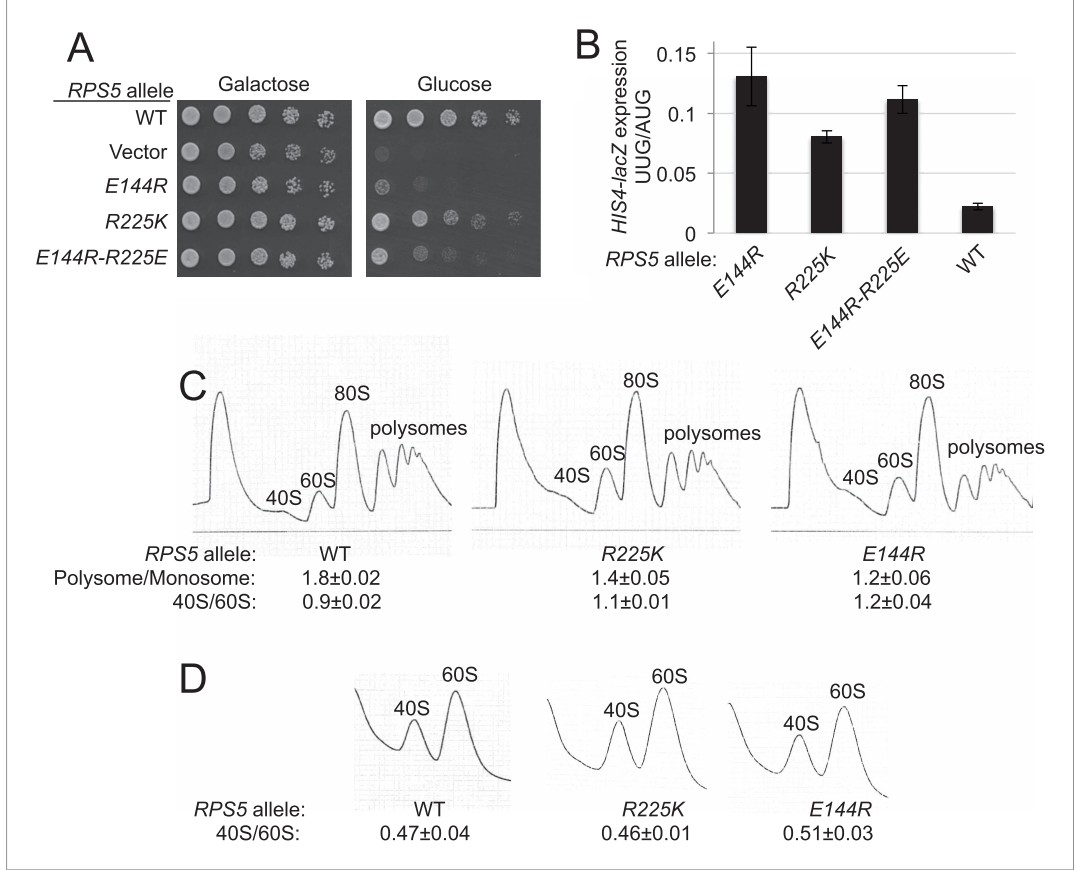

**Figure 3**. *RPS5* mutations *E144R* and *R225K* impair translation initiation and elevate UUG initiation without reducing 40S subunit abundance in vivo. (**A**) 10-fold serial dilutions of transformants of $P_{GAL1}$-*RPS5 his4-301* strain (JVY07) with the indicated plasmid-borne *RPS5* alleles were spotted on synthetic medium supplemented with histidine and containing galactose (SGal + His + Ura + Trp) or glucose (SD + His + Ura + Trp) as carbon source and incubated at 30℃ for 3 days (Glucose) or 4 days (Galactose). (**B**) Strains from (**A**) also harboring *HIS4-lacZ* reporters with AUG or UUG start codons (plasmids p367 and p391, respectively) were cultured in SD + His + Trp at 30℃ to $A_{600}$ of ~1 and β-galactosidase specific activities were measured in WCEs in units of nanomoles of o-nitrophenyl- β-D-galactopyranoside (ONPG) cleaved per min per mg of total protein. Ratios of mean expression of the UUG and AUG reporters calculated from four transformants are plotted with error bars (indicating S.E.M.s). (**C**) Strains from (**A**) were cultured in SD + His + Ura + Trp at 30℃ to $A_{600}$ of ~1, and cycloheximide was added prior to harvesting. WCEs were separated by sucrose density gradient centrifugation and scanned at 254 nm to yield the tracings shown. Mean Polysome/Monosome and 40S/60S ratios (and S.E.M.s) from four replicates are indicated. Student's *t*-test indicates that the mean values for polysome/monosome *ratio* in the *RPS5* mutants are reduced significantly from the WT (p < 0.0005). (**D**) Similar to (**C**) but the cultures were not treated with cycloheximide and lysed in buffers without MgCl₂ to allow separation of the dissociated ribosomal subunits.

(*E144R*) with the stronger Slg⁻ phenotype (*Figure 3A*). Neither mutant significantly perturbed the ratio of 40S to 60S subunits (*Figure 3C,D*), suggesting that the initiation defects arise from altered 40S function rather than abnormalities in expression of Rps5, 40S biogenesis, or stability of mature 40S subunits. Thus, it appears that *E144R* and *R225K* reduce the function of Rps5 in stimulating the rate of general translation initiation and promoting accurate start codon selection.

### *E144R* and *R225K* elevate UUG initiation indirectly by exacerbating the effect of poor context of the *SUI1* start codon and thereby reducing eIF1 abundance

In addition to increasing initiation from near-cognate UUG codons, certain Sui⁻ mutations in eIF1, eIF1A, and eIF2β are known to enhance initiation from AUG codons in poor context. As such, they

suppress the effects of the suboptimal context of the AUG codon of *SUI1* mRNA and increase expression of the encoded eIF1 protein (*Martin-Marcos et al., 2011*). This phenotype is illustrated for the Sui⁻ *sui1-L96P* mutation in *Figure 4A* (lanes 7, 8 vs 5, 6). However, unlike previously described

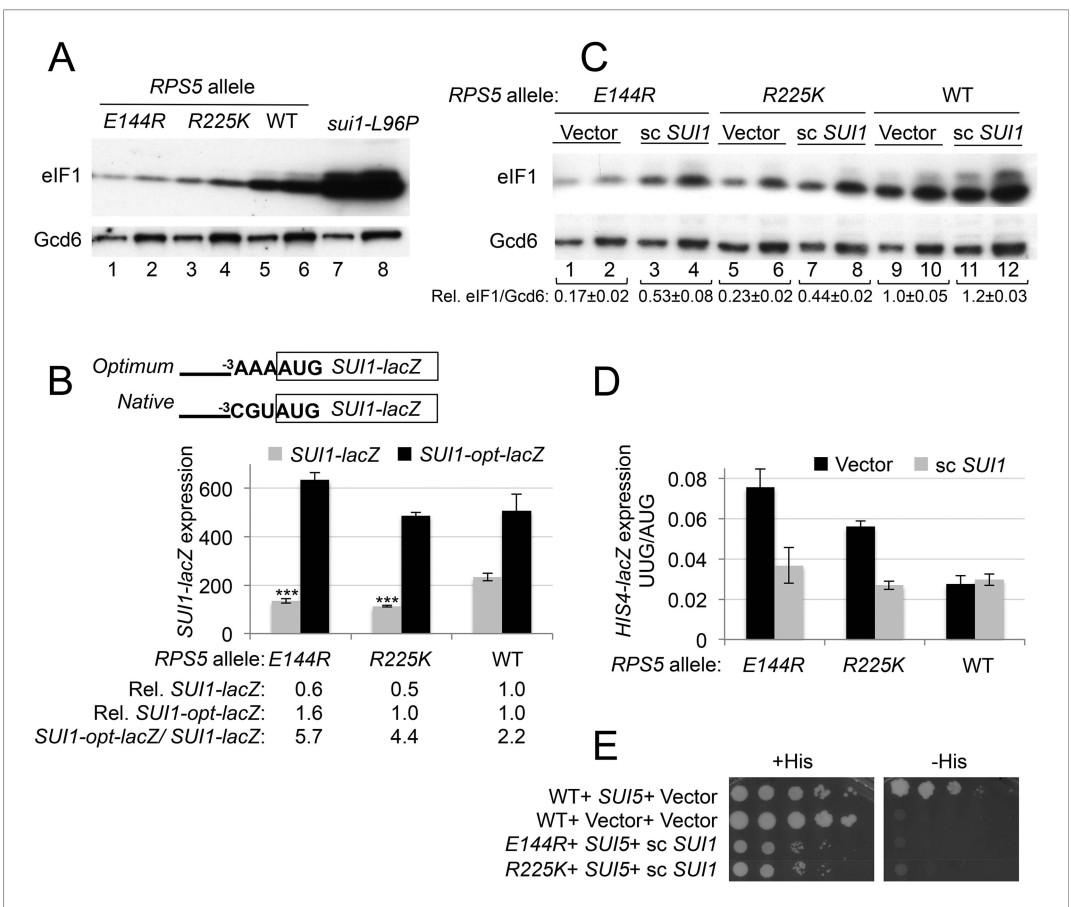

**Figure 4**. *RPS5* mutations *E144R* and *R225K* exacerbate poor context at the native *SUI1* AUG to reduce eIF1 expression and indirectly confer Sui⁻ phenotypes, but evoke Ssu⁻ phenotypes when eIF1 abundance is boosted. (**A**) WCEs of strains from *Figure 3A*, and from *sui1-L96P* strain H4564, were subjected to Western analysis using antibodies against eIF1 or Gcd6 (as loading control). Two amounts of each extract differing by a factor of two were loaded in successive lanes. (**B**) Strains from *Figure 3A* also harboring *SUI1-lacZ* (pPMB24) or *SUI1- opt-lacZ* (pPMB25) reporters were cultured and assayed for β-galactosidase activities as described in *Figure 3B*. Mean expression levels and S.E.M.s from four transformants are plotted, and relative (Rel.) mean expression levels normalized to that of the WT strain are listed below the histogram. Student's *t*-test indicates that the mean values for *SUI1-lacZ* expression in the *RPS5* mutants are reduced significantly from the WT (***$p < 0.0005$). (**C**) WCEs of strains from *Figure 3A* also harboring sc *SUI1* plasmid pPMB21 or empty vector were subjected to Western analysis as in (**A**). Signal intensities were quantified and mean eIF1/Gcd6 ratios are listed below the blot with S.E.Ms (**D**) *HIS4-lacZ* reporters with AUG or UUG start codons were assayed in strains from (**C**) as in *Figure 3B*. (**E**) *his4-301* strains with the indicated WT or mutant *RPS5* alleles (from *Figure 3A*) harboring sc *SUI1* plasmid pPMB21, *SUI5* plasmid p4281, or empty vectors were spotted on SD plates containing (SD + His) or lacking histidine (SD-His) and incubated for 3 days and 5 days, respectively.

The following figure supplements are available for figure 4:

**Figure supplement 1**. Increased *SUI1* gene dosage partially rescues the Slg⁻ phenotype of *RPS5* mutations *E144R* and *R225K*.

**Figure supplement 2**. *RPS5* mutation *E144R* confers a Gcd⁻ phenotype, derepressing *GCN4-lacZ* expression with restored eIF1 expression.

mutations that enhance UUG recognition, *rps5-E144R* and *-R225K* paradoxically decrease eIF1 abundance (*Figure 4A*, lanes 1–4 vs 5, 6) to a degree that correlates with their elevated UUG:AUG *HIS4-lacZ* initiation ratios (*Figure 3B*). Consistently, *E144R* and *R225K* reduce expression of a *SUI1-lacZ* reporter bearing the native, suboptimal context at the three nucleotides preceding the AUG codon ($^{-3}$CGU$^{-1}$), but not that of a modified *SUI1$_{opt}$-lacZ* reporter with an optimized AUG context ($^{-3}$AAA$^{-1}$) (*Figure 4B*). Thus, the *rps5* mutations exacerbate the effect of suboptimal context and decrease AUG recognition on WT *SUI1* mRNA. The reduction in eIF1 abundance implies that the *rps5* mutations overcome the autoregulation of eIF1 expression, wherein low eIF1 levels suppress the effect of poor context at the *SUI1* AUG codon to boost eIF1 abundance (*Ivanov et al., 2010*; *Martin-Marcos et al., 2011*). Accordingly, it appears that the *rps5* mutations evoke a pronounced defect in recognition of the native *SUI1* AUG codon that prevails even at low cellular concentrations of eIF1 that would normally boost *SUI1* translation.

Interestingly, the discrimination against native poor context at *SUI1* with attendant reduced eIF1 expression represents a hyperaccuracy phenotype displayed by known Ssu$^-$ (<u>S</u>uppressor of <u>Sui</u>$^-$) mutations in eIF1, eIF1A, and eIF2β, which additionally suppress the elevated UUG:AUG ratio conferred by various Sui$^-$ mutations (*Martin-Marcos et al., 2011*). Hence, the fact that *rps5-E144R* and *-R225K* discriminate against poor context at *SUI1* but elevate the UUG:AUG ratio seems paradoxical. However, their elevated UUG:AUG ratio (hypoaccurate) phenotype could be explained by the reduced levels of eIF1 present in these *rps5* mutants (*Hinnebusch, 2011*). Indeed, we found that increasing the level of WT eIF1 by adding an extra plasmid-borne copy of WT *SUI1* (*Figure 4C*, sc*SUI1* vs Vector transformants) mitigated the Sui$^-$ phenotypes of both *rps5* mutants, reducing their UUG:AUG ratios to essentially WT levels (*Figure 4D*). Introducing sc*SUI1* also mitigated their Slg$^-$ phenotypes, slightly for *-R225K* and substantially for *-E144R* (*Figure 4—figure supplement 1*). The resulting *rps5/scSUI1* strains exhibit reduced eIF1 levels compared to WT cells containing an extra copy of *SUI1* (*Figure 4C*, lanes 3–4 and 7–8 vs 11–12), indicating that the *rps5* mutations still exacerbate the effect of poor AUG context at higher levels of eIF1 expression. We conclude that the increased recognition of UUG start codons conferred by the *rps5* mutations is an indirect consequence of their reduced expression of eIF1.

As noted above, reducing eIF1 expression by discriminating against the poor context of the *SUI1* AUG codon is a phenotype of known Ssu$^-$ (hyperaccuracy) mutants (*Martin-Marcos et al., 2011*). Since *rps5* mutations also reduced eIF1 expression in a context-dependent manner, we next examined whether they exhibit Ssu$^-$ phenotypes by testing them for the ability to suppress the dominant Sui$^-$ phenotype of the *SUI5* variant of eIF5 (eIF5-G31R) (*Huang et al., 1997*). This test was conducted using the *RPS5* mutant strains harboring sc*SUI1* to compensate for the reduced eIF1 expression responsible for their Sui$^-$ phenotypes. Introducing *SUI5* on a plasmid conferred the expected His$^+$/Sui$^-$ phenotype in the *his4-301* strain harboring WT *RPS5* and native eIF1 levels (*Figure 4E*, compare row 1 with 2). Importantly, this His$^+$ phenotype was eliminated in the corresponding *rps5-E144R* and *-R225K* mutants when eIF1 levels were boosted by introduction of sc*SUI1*, as the reduction in growth on −His medium was greater than that seen on +His medium in the *rps5/SUI5* mutants vs the *RPS5/SUI5* strain (*Figure 4E*, rows 3–4 vs 1). Suppression of the His$^+$ phenotype of *SUI5* could arise from defective induction of *GCN4* mRNA with attendant impairment of *HIS4* transcription (*Hinnebusch, 2005*); however, at least *rps5-E144R* does not reduce the expression of a *GCN4-lacZ* reporter in amino acid starved cells containing an extra copy of *SUI1* (*Figure 4—figure supplement 2*). Hence, similar to known Ssu$^-$ mutations in eIF1 or eIF1A (*Martin-Marcos et al., 2011*), the *rps5-E144R* mutation appears to suppress recognition of the UUG start codon of *his4-301* mRNA in addition to discriminating against poor context at the *SUI1* AUG codon.

## *E144R* and *R225K* confer leaky scanning of an upstream AUG start codon

As noted above, the *rps5* mutations decrease recognition of the *SUI1* start codon and suppress UUG initiation when eIF1 levels are restored. We next asked whether they also decrease recognition of an upstream AUG codon and allow leaky scanning to the downstream ORF. A *GCN4-lacZ* reporter was employed with a modified version of upstream ORF1 that is elongated to overlap the *GCN4* ORF (el.uORF1). This construct is ideally suited for this query because virtually all scanning ribosomes normally recognize the uORF1 AUG (uAUG-1), and reinitiation at the downstream AUG of the *GCN4* main ORF following termination of el.uORF1 translation is almost non-existent, so that translation of

the main ORF is extremely low (*Grant et al., 1994*). Remarkably, *rps5-E144R* confers a dramatic increase in leaky scanning through el.uORF1, elevating *GCN4-lacZ* expression by 20-fold for the construct containing optimum context ($^{-3}$AAA$^{-1}$) at uAUG-1 (*Figure 5*, row 1, WT vs *E144R*). This effect is nearly comparable to the ~40-fold increase in leaky scanning seen in WT cells for the extremely weak uAUG-1 context of $^{-3}$UUU$^{-1}$ (*Figure 5*, WT, row 5 vs row 1). *rps5-E144R* also evokes a large ~10-fold increase in leaky scanning for the el.uORF1 construct with the uAUG-1 context of intermediate strength ($^{-3}$UAA$^{-1}$), but only a ~fourfold increase with the weakest context of $^{-3}$UUU$^{-1}$ (*Figure 5*, rows 3 and 5, WT vs *E144R*). Expression of the construct lacking uAUG-1 is not significantly affected by *rps5-E144R* (*Figure 5*, row 7, WT vs *E144R*), consistent with leaky scanning being the source of elevated *GCN4-lacZ* expression for the various el.uORF1 constructs. The *R225K* mutation also increases leaky scanning of uAUG-1, but to a lesser degree: 3.5-fold for $^{-3}$AAA$^{-1}$, 3.3-fold for $^{-3}$UAA$^{-1}$, and 1.7-fold for $^{-3}$UUU$^{-1}$ (*Figure 5*, rows 1, 3, 5; WT vs *R225K*). The increases in leaky scanning conferred by the *rps5* mutations were relatively unaffected by the restoration of WT eIF1 levels by introducing sc*SUI1* (*Figure 5*, rows 2, 4, 6 vs 1, 3, 5, respectively). This was anticipated because the reduced levels of eIF1 in the *rps5* mutants (lacking sc*SUI1*) would not be expected to reduce uAUG-1 recognition and confer leaky scanning, as decreased eIF1 abundance is associated with increased start codon recognition (at least for near-cognate start codons or AUG codons in poor context) (*Pestova and Kolupaeva, 2002*; *Ivanov et al., 2010*; *Martin-Marcos et al., 2011*). We conclude that the *rps5* substitutions impair recognition of *GCN4* uAUG-1, whether located in perfect or poor surrounding sequence context, to allow increased translation of the downstream *GCN4* coding sequences.

## Ssu$^{-}$ substitution *E144R* destabilizes the P$_{IN}$ conformation of the 48S PIC in vitro

The multiple defects in start codon recognition conferred by *rps5-E144R* suggest that it destabilizes the P$_{IN}$ state of the 48S PIC. We tested this hypothesis by analyzing the effects of *E144R* on the equilibrium and rate constants governing TC binding to the 40S subunit in the yeast reconstituted translation system. To this end, we purified 40S subunits from *rps5Δ::kanMX* deletion strains harboring either plasmid-borne *rps5-E144R* or WT *RPS5* as the only source of Rps5. We began by measuring the affinity of WT TC, assembled with [$^{35}$S]-Met-tRNA$_i$, for mutant or WT 40S subunits in the presence of saturating eIF1, eIF1A and a model mRNA containing an AUG start codon (mRNA(AUG)), using native gel electrophoresis to separate 40S-bound and unbound fractions of TC. Reactions conducted with increasing concentrations of 40S subunits revealed that 43S•mRNA(AUG) complexes assembled with either *E144R* or WT 40S subunits have relatively high affinities for TC (*Figure 6A*), with K$_d$ values of ≤1 nM (*Figure 6E*). In the absence of mRNA, the affinities for TC are similar between 43S PICs assembled with mutant or WT 40S subunits (*Figure 6E*); however, the endpoint of the reaction is

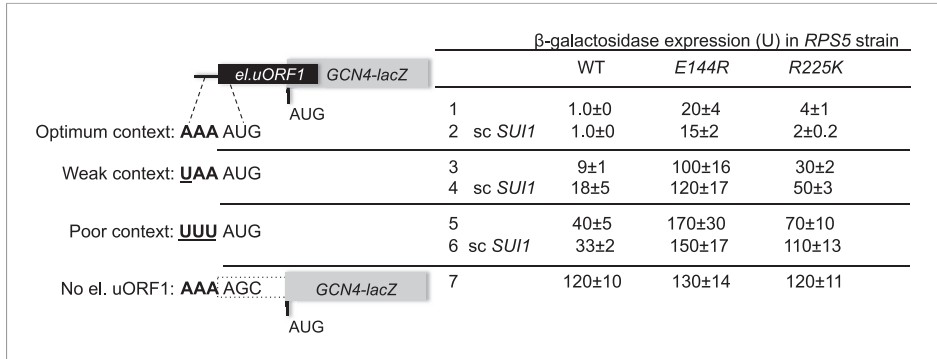

| | | | β-galactosidase expression (U) in *RPS5* strain | | |
| | | | WT | E144R | R225K |
|---|---|---|---|---|---|
| Optimum context: **AAA** AUG | 1 | | 1.0±0 | 20±4 | 4±1 |
| | 2 | sc *SUI1* | 1.0±0 | 15±2 | 2±0.2 |
| Weak context: **UAA** AUG | 3 | | 9±1 | 100±16 | 30±2 |
| | 4 | sc *SUI1* | 18±5 | 120±17 | 50±3 |
| Poor context: **UUU** AUG | 5 | | 40±5 | 170±30 | 70±10 |
| | 6 | sc *SUI1* | 33±2 | 150±17 | 110±13 |
| No el. uORF1: **AAA** AGC | 7 | | 120±10 | 130±14 | 120±11 |

**Figure 5**. *RPS5* mutations *E144R* and *R225K* confer strong leaky scanning of *GCN4* uAUG-1 in vivo. β-galactosidase activities were measured in WCEs of strains from *Figure 4C* harboring the sc*SUI1* plasmid (as indicated) and el.uORF1 *GCN4-lacZ* reporters pC3502, pC4466, or pC3503 containing, respectively, the depicted optimum, weak, or poor context of uAUG-1; or uORF-less *GCN4-lacZ* reporter pC3505 with a mutated uAUG-1. Mean expression values with S.E.M.s were determined from four transformants as described in *Figure 3B*.

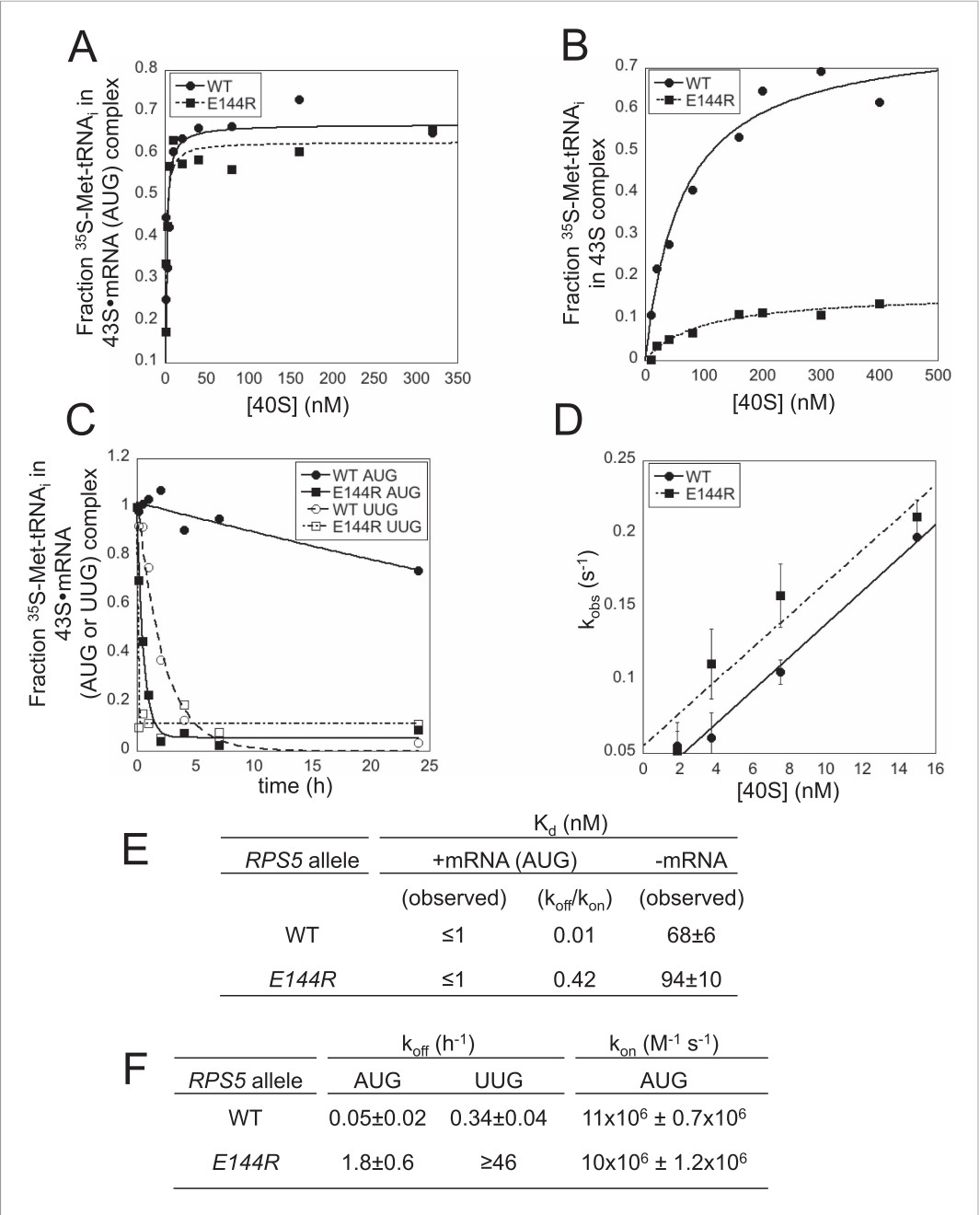

Figure 6. Rps5 Ssu⁻ substitution E144R destabilizes the P_IN state in vitro to a greater extent at UUG vs AUG start codons. (**A**, **B**) Determination of $K_d$ values for TC with [³⁵S]-Met-tRNA_i binding to 40S·eIF1·eIF1A complexes assembled with WT or *E144R* mutant 40S subunits and either mRNA (AUG) (**A**) or without mRNA (**B**). (**C**) Analysis of TC dissociation from 43S·mRNA complexes assembled with WT or *E144R* mutant 40S subunits and either mRNA (AUG) or mRNA(UUG). Representative curves selected from at least three independent experiments are shown. (**D**) Determination of $k_{on}$ values for TC binding to 40S·eIF1·eIF1A complexes from plots of observed rate constants ($k_{obs}$) vs 40S concentration for WT or *E144R* mutant 40S subunits and mRNA(AUG). (**E**, **F**) $K_d$, $k_{off}$ and $k_{on}$ values with S.E.M.s determined in (**A–D**).

The following figure supplement is available for figure 6:

**Figure supplement 1**. *RPS5* mutation *E144R* confers a Gcd⁻ phenotype, derepressing *GCN4-lacZ* expression.

markedly reduced for the *E144R* complexes (*Figure 6B*). It was previously proposed that the endpoints of TC binding reactions achieved at saturating 40S concentrations reflect the distribution of PICs between open and closed states. The open state was assumed to be unstable during electrophoresis, and thus could not be visualized, leading to endpoints of <1 (measured as fractions of TC bound to 40S complexes) for reactions using mRNA lacking an AUG codon (*Kapp et al., 2006*; *Kolitz et al., 2009*), or tRNA$_i^{Met}$ mutants (*Dong et al., 2014*), in which the open complex is favored over the closed state. Hence, the reduced endpoint seen in *Figure 6B* suggests that the closed state of 43S complexes formed with *E144R* mutant ribosomes is unstable and rearranges to the less stable, open conformation during electrophoresis. This interpretation supports the hypothesis that *E144R* destabilizes the closed state of the PIC.

Direct evidence for this last conclusion came from determining rate constants for TC association and dissociation for 43S complexes bound to mRNA. To measure the TC off-rate ($k_{off}$), 43S•mRNA complexes were formed as above using TC assembled with [$^{35}$S]-Met-tRNA$_i$, and the amount of [$^{35}$S]-Met-tRNA$_i$ remaining in the slowly-migrating PIC was measured at different times after adding a chase of excess unlabeled TC. In agreement with previous findings (*Kolitz et al., 2009*; *Dong et al., 2014*; *Martin-Marcos et al., 2014*), TC dissociates very little from WT PICs formed with mRNA(AUG) over the time course of the experiment, yielding a rate constant of only 0.05 hr$^{-1}$ (*Figure 6C*; summarized in *Figure 6F*). By contrast, TC dissociation from WT PICs assembled on an otherwise identical mRNA containing a UUG start codon is ~sevenfold faster ($k_{off}$ = 0.34 hr$^{-1}$), reflecting the reduced stability of the P$_{IN}$ state at this near-cognate start codon (*Figure 6C,F*) (*Kolitz et al., 2009*). Remarkably, for 43S•mRNA(AUG) complexes assembled with *E144R* 40S subunits, the dissociation rate was increased ~36-fold compared to that seen for the corresponding WT complexes (from 0.05 hr$^{-1}$ to 1.8 hr$^{-1}$; *Figure 6C,F*). An even larger increase in $k_{off}$ of ~130-fold was measured for mRNA (UUG) complexes assembled with *E144R* vs WT 40S subunits (≥46 hr$^{-1}$ vs 0.34 hr$^{-1}$; *Figure 6C,F*). These findings provide strong biochemical evidence that *E144R* destabilizes P$_{IN}$ at both AUG and UUG start codons with a relatively stronger effect on the near-cognate triplet, which coincides with the in vivo effects of *E144R* of reducing recognition of the *SUI1* AUG and *GCN4* uAUG-1 start codons, and of suppressing UUG initiation on *his4-301* mRNA.

The rates of TC association ($k_{on}$) were measured by mixing labeled TC with different concentrations of WT or *E144R* 40S subunits and saturating eIF1, eIF1A and mRNA(AUG). Aliquots were removed at different time points, the reactions terminated with excess unlabeled TC, and the amount of labeled TC in PICs was measured by native gel electrophoresis. The slope of the plot of the pseudo-first-order rate constants ($k_{obs}$) for PIC formation vs 40S concentration yields the second-order rate constant ($k_{on}$) (*Kolitz et al., 2009*). The $k_{on}$ values measured for WT and *E144R* 40S subunits were essentially identical (*Figure 6D,F*), indicating that PICs formed with the mutant ribosomes assemble the P$_{OUT}$ complex and rearrange to P$_{IN}$ at the same rates achieved with WT ribosomes, and that *rps-E144R* primarily reduces the stability of the P$_{IN}$ state. Calculation of $K_d$ values using the measured rate constants $k_{off}$ and $k_{on}$ reveals that *E144R* decreases the affinity of TC for 43S•mRNA(AUG) complexes by ~40-fold (*Figure 6E*, $k_{off}/k_{on}$). Together, the in vitro experiments demonstrate that *E144R* reduces the affinity of TC for 43S•mRNA PICs by destabilizing the P$_{IN}$ state, with a relatively greater effect at UUG vs AUG start codons.

Interestingly, we found that *E144R* confers the Gcd$^-$ phenotype, derepressing a *GCN4-lacZ* reporter by more than fourfold in non-starvation conditions (*Figure 6—figure supplement 1*), which indicates a defect in TC binding to 40S subunits in vivo. A decreased rate of TC binding derepresses *GCN4-lacZ* expression because scanning 40S subunits that have translated uORF1 and resumed scanning can bypass the start codons of the inhibitory uORFs 2–4 before rebinding TC, and then reinitiate further downstream at the *GCN4* AUG codon (*Hinnebusch, 2005*). Derepression of *GCN4-lacZ* by *E144R* was evident even in the presence of sc*SUI1* (*Figure 4—figure supplement 2*, *E144R*/sc*SUI1* vs WT, unstarved), indicating that it does not result solely from the reduced eIF1 abundance in this mutant. Because *E144R* does not reduce the rate of TC binding to 43S·mRNA complexes (~WT $k_{on}$ value, *Figure 6F*), but greatly increases its off-rate (elevated $k_{off}$, *Figure 6F*), we infer that the Gcd$^-$ phenotype of *E144R* arises instead from dissociation of TC from a fraction of re-scanning 40S subunits, enabling their bypass of uORFs 2–4, followed by re-binding of TC in time to reinitiate at *GCN4*. This mechanism was described previously for Gcd$^-$ substitutions of 18S rRNA residues in the P site of the 40S subunit that, like *rps5-E144R*, destabilize TC binding in vitro and confer leaky-scanning of an uAUG in vivo (*Dong et al., 2008*).

## Substitutions in the loop region of the Rps5 β-hairpin increase fidelity of start codon selection independently of context nucleotides

Having concluded that *rps5-E144R* can suppress UUG initiation once native eIF1 levels have been restored, we examined the remaining β-hairpin substitutions we constructed for this Ssu⁻ phenotype. Remarkably, eight different mutations affecting various loop residues were found to suppress the His⁺ phenotypes conferred by dominant Sui⁻ alleles *SUI3-2* and *SUI5,* including *R148A/E, R156A/E, R157A/E, A154R* and *A155E* (*Figure 7A,* -His panel; cf. WT and *rps5* strains harboring *SUI3-2* or *SUI5*). With the possible exception of *R157E,* they also suppressed the dominant Slg⁻ phenotype of *SUI5* (*Figure 7A,* +His panel; cf. WT and *rps5* strains harboring *SUI5*)—a hallmark of known Ssu⁻ mutations in eIF1 (*Martin-Marcos et al., 2014*). Furthermore, *R148A/E, R156A/E* and *R157A* suppressed the elevated UUG:AUG ratio of *HIS4-lacZ* expression conferred by *SUI3-2* (*Figure 7B,C*), demonstrating *bona fide* Ssu⁻ phenotypes for these mutations.

In addition to discriminating against non-AUG codons, Ssu⁻ mutations in eIF1 and eIF1A exacerbate the effect of suboptimal context of the *SUI1* AUG start codon and reduce eIF1 expression. As such, they exacerbate the differential expression of *SUI1-lacZ* fusions containing native, suboptimal context vs optimized context by specifically reducing expression of the native-context reporter (*Martin-Marcos et al., 2011*). However, none of the *rps5* Ssu⁻ mutants exhibit diminished eIF1 abundance (*Figure 7D,E*), or selectively diminish expression of the *SUI1-lacZ* fusion with native context (*Figure 7F*). Nor do they increase leaky scanning of uAUG-1 regardless of its context in the el. uORF1 reporters (*Figure 7—figure supplement 1*). Thus, unlike known Ssu⁻ mutations affecting eIF1 and eIF1A, the *rps5* Ssu⁻ substitutions in the β-hairpin loop suppress recognition of UUG codons without affecting utilization of AUG codons in poor context.

## Ssu⁻ substitution *R148E* destabilizes the $P_{IN}$ conformation of the PIC at UUG codons

To reveal the molecular mechanism of the Ssu⁻ substitutions in the β-hairpin loop, we analyzed mutant 40S subunits purified from *rps5-R148E* cells in the reconstituted yeast system. Measurements of TC binding to 43S·mRNA(AUG) complexes or 43S complexes without mRNA revealed reaction endpoints (*Figure 8A,B*) and $K_d$ values (<1 nM for 43S·mRNA(AUG) complexes, *Figure 8E*) indistinguishable between WT and *R148E* mutant ribosomes, as were rates of TC dissociation ($k_{off}$) from these complexes containing AUG start codons (*Figure 8C*, AUG, WT vs R148E; *Figure 8F*, WT eIF2β, AUG values). However, *R148E* increased the $k_{off}$ for 43S·mRNA(UUG) complexes by ~twofold, suggesting destabilization of $P_{IN}$ specifically at UUG codons (*Figure 8C,F*, WT eIF2β, UUG). To support this conclusion, we repeated the $k_{off}$ measurements using eIF2 harboring the Sui⁻ substitution in eIF2β encoded by *SUI3-2* (S264Y). Consistent with previous results (*Martin-Marcos et al., 2014*), in reactions with WT 40S subunits, *SUI3-2* eliminates detectable TC dissociation from AUG complexes and also delays TC dissociation from UUG complexes (*Figure 8D*) compared to that seen using WT eIF2 (*Figure 8C*), thus decreasing the $k_{off}$ for UUG complexes by ~threefold (*Figure 8F*; WT eIF2β vs eIF2β-S264Y, WT *RPS5,* UUG complexes). These results are consistent with the elevated UUG initiation conferred by *SUI3-2* in vivo. Importantly, in assays with the *SUI3-2* variant of eIF2, *rps5-R148E* produced a marked, ~fourfold increase in $k_{off}$ for the UUG complexes without affecting dissociation of the corresponding AUG complexes (*Figure 8D,F*). Thus, *rps5-R148E* preferentially destabilizes the $P_{IN}$ conformation at UUG start codons, overriding the opposing effect of *SUI3-2* of enhancing the stability of the UUG complex. These biochemical results are in accordance with our finding that *rps5-R148E* suppresses the elevated UUG:AUG initiation ratio conferred by *SUI3-2* in vivo.

## Discussion

In this study, we obtained genetic and biochemical evidence implicating the β-hairpin of Rps5 in achieving efficient and accurate start codon recognition in vivo. In the recent py48S cryo-EM structure (*Hussain et al., 2014*), this domain projects into the mRNA exit channel of the 40S subunit and the hairpin loop approaches the key context nucleotide at the −3 position of mRNA. The β-hairpin also interacts with eIF2α-DI, which mimics an E-site tRNA and contacts the Met-tRNA_i in the P site (*Figure 2A*). Our genetic findings indicate that the E144R substitution in β-strand 1 of the hairpin reduces the rate of bulk translation initiation (*Figure 3C*) and dramatically impairs recognition of *GCN4* uAUG-1 in optimal context by the scanning PIC, conferring a higher incidence of leaky scanning

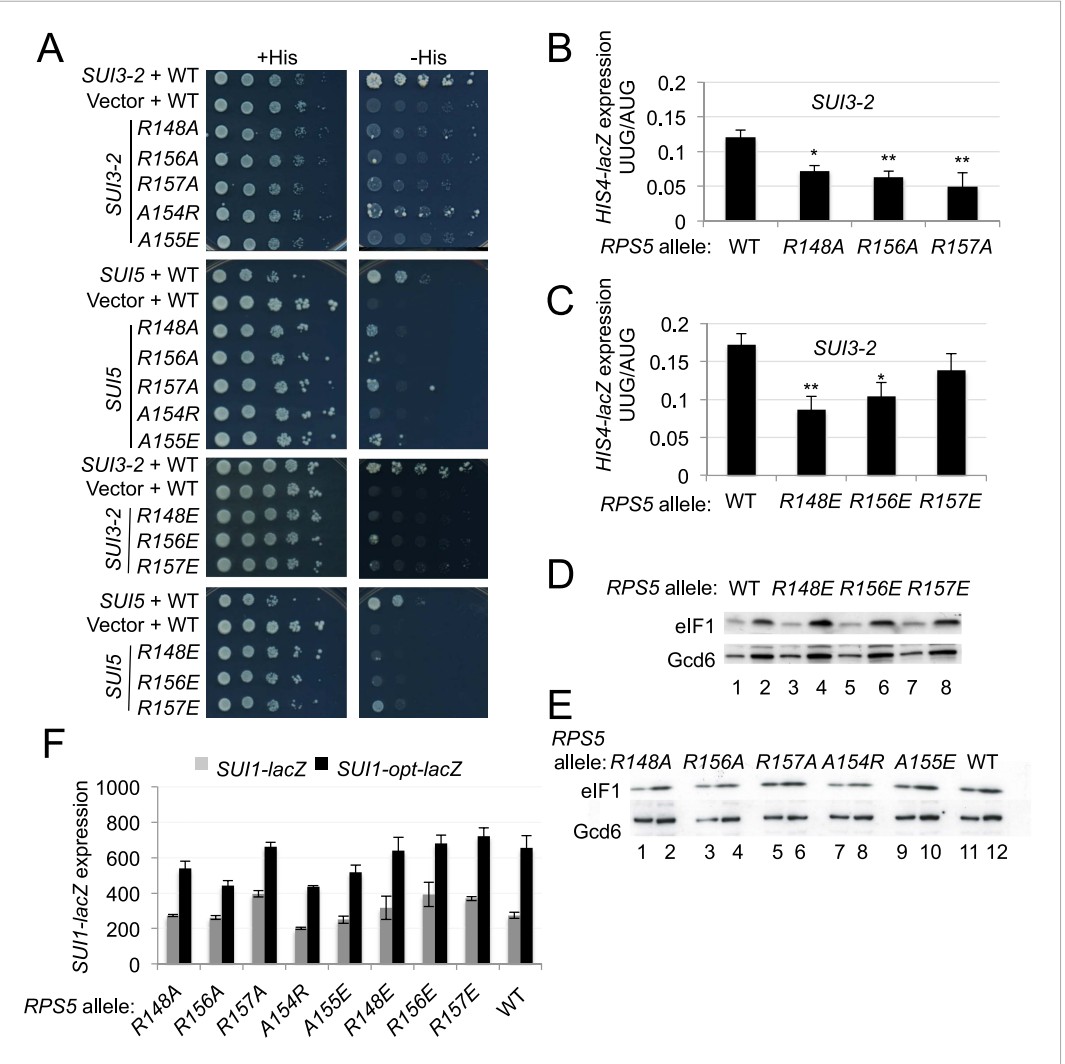

**Figure 7**. Substitutions in loop residues of the Rps5 β-hairpin confer Ssu⁻ phenotypes. (**A**) 10-fold serial dilutions of $P_{GAL1}$-*RPS5 his4-301* strain (JVY07) transformed with the indicated plasmid-borne *RPS5* alleles and either *SUI3-2* plasmid p4280, *SUI5* plasmid p4281, or empty vector were spotted on SD + His + Ura (+His) or SD + Ura (−His) and incubated at 30°C for 3 days and 5 days, respectively. (**B**, **C**) Strains from (**A**) also harboring *HIS4-lacZ* reporters with AUG or UUG start codons (plasmids p367 and p391, respectively) were analyzed as in *Figure 3B*. Ratios of mean expression of the UUG and AUG reporters calculated from four transformants are plotted with S.E.M.s. Student's *t*-test indicates that the mean UUG/AUG expression in the *RPS5* mutants is significantly reduced when compared to WT (*p < 0.05, **p < 0.005). (**D**, **E**) WCEs of *his4-301* strains with the indicated *RPS5* alleles were subjected to Western analysis as in *Figure 4A*. (**F**) WCEs of strains from (**D**, **E**) also harboring *SUI1-lacZ* (pPMB24) or *SUI1-opt-lacZ* (pPMB25) reporters were assayed for β-galactosidase activities as described in *Figure 4B*. Mean expression levels and S.E.M.s from four transformants are plotted.

The following figure supplement is available for figure 7:

**Figure supplement 1**. Substitutions in the loop of the Rps5 β-hairpin do not increase leaky scanning of *GCN4* uAUG-1.

for the *el.uORF1-GCN4-lacZ* reporter than described thus far for any initiation factor mutation (*Elantak et al., 2010*). The E144R mutation also impairs recognition of the *SUI1* AUG codon in its native, suboptimal context, and suppresses utilization of the UUG start codon in *his4-301* mRNA in different Sui⁻ mutants to confer an Ssu⁻ phenotype. Our biochemical analysis of E144R mutant 40S subunits revealed a drastic destabilization of the $P_{IN}$ state of reconstituted 48S PICs at AUG or UUG

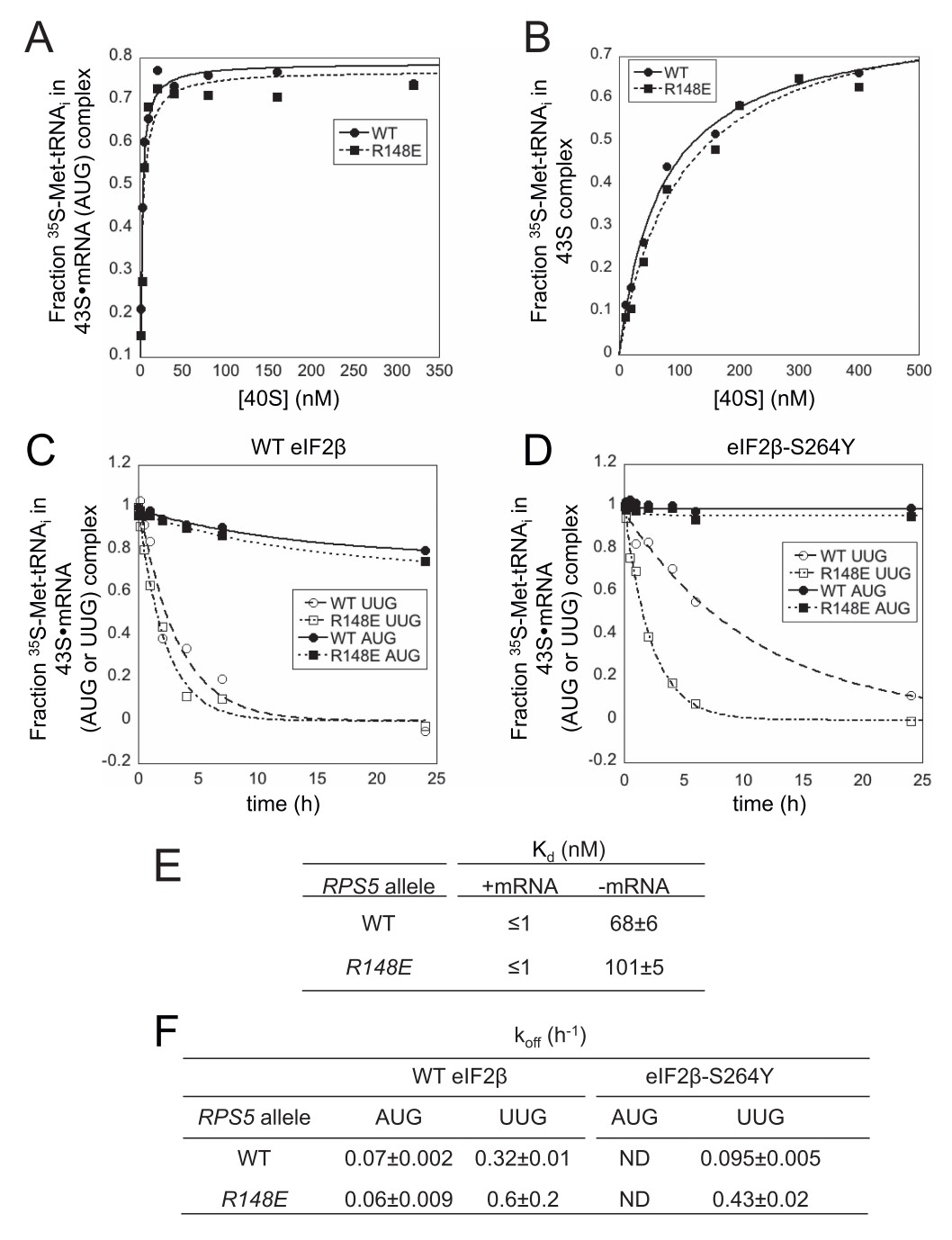

**Figure 8.** Rps5 Ssu[-] substitution R148E destabilizes $P_{IN}$ in vitro selectively at UUG codons. (**A**, **B**) Determination of $K_d$ values for TC with [35S]-Met-tRNA$_i$ binding to 40S·eIF1·eIF1A complexes assembled with WT or *R148E* mutant 40S subunits and either mRNA (AUG) (**A**) or without mRNA (**B**). (**C**, **D**) Analysis of TC dissociation from 43S·mRNA complexes assembled with WT or *R148E* mutant 40S subunits and mRNA(AUG) or mRNA(UUG), conducted using WT eIF2 (**C**) or eIF2β-S264Y mutant eIF2 (**D**). Representative curves selected from at least three independent experiments are shown. (**E**, **F**) $K_d$, $k_{off}$ values with S.E.M.s determined in (**A**–**D**). ND, no dissociation observed.

codons (*Figure 9*), with a stronger effect on the inherently less stable UUG complexes. These biochemical phenotypes can account for both the defects in AUG recognition and the reduction in UUG:AUG initiation ratio (Ssu⁻ phenotype) conferred by *rps5-E144R* in vivo.

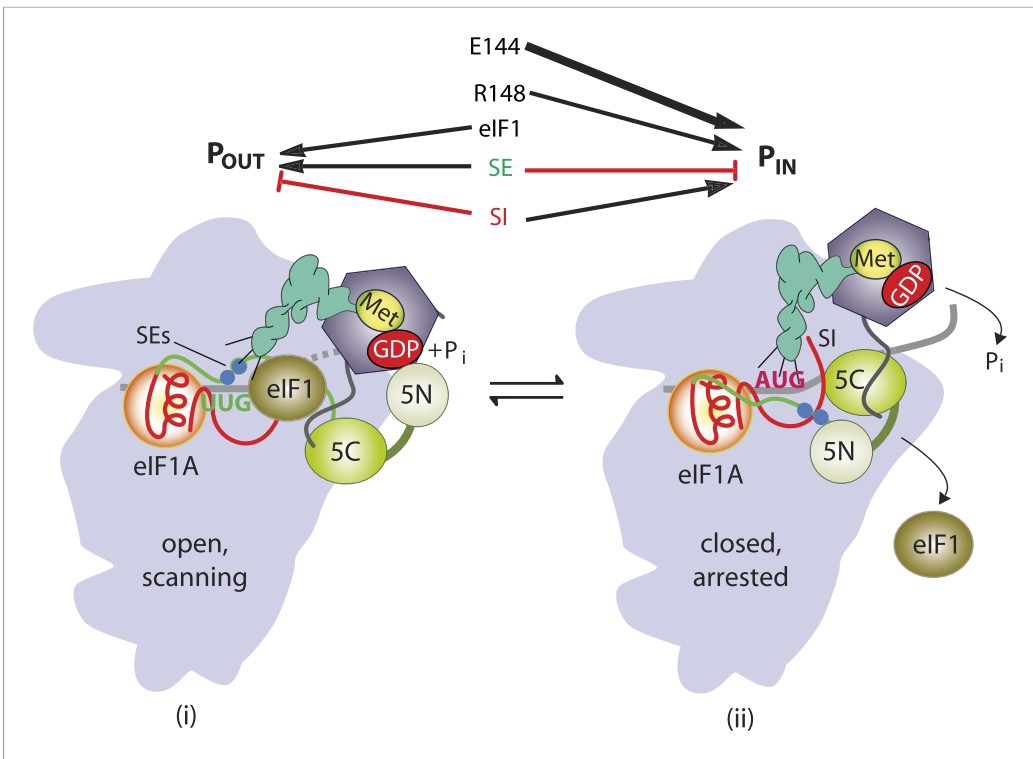

**Figure 9**. β-hairpin of Rps5 has a critical role in start codon recognition during translation initiation by stabilizing initiator tRNA binding to the pre-initiation complex. Model summarizing the role of the conserved β-hairpin residues in Rps5 in start codon recognition. See **Figure 1** for description of the open/$P_{OUT}$ and closed/$P_{IN}$ states of the PIC and roles of eIF1 and the SE/SI elements of eIF1A in regulating conformational rearrangements and reactions accompanying AUG recognition. Results from this study indicate that Rps5 β-hairpin residues E144 and R148 function in stabilizing the $P_{IN}$ conformation of TC binding, with E144 having a stronger effect, as indicated by the thicker arrow.

The following figure supplement is available for figure 9:

**Figure supplement 1**. Rps5 β-hairpin loop is in proximity to rRNA helix 23.

The *E144R* mutation conferred a greater increase in leaky scanning of *GCN4* uAUG-1 in optimum context (~20-fold) vs uAUG-1 in poor context (~fourfold). This difference could be interpreted to indicate that *E144R* disrupts recognition of optimum context nucleotides, so that its deleterious effect on AUG recognition is dampened when the optimum context nucleotides are absent. However, this interpretation would overlook the fact that it is impossible to detect an increase in leaky scanning of uAUG-1 in poor context by 20-fold, as only a ~threefold reduction in its recognition yields *GCN4-lacZ* expression essentially indistinguishable from that seen in the absence of uAUG-1 itself (*Figure 5*, cf. constructs UUUAUG vs AAAAGC in WT cells). Thus, it seems plausible that *E144R* suppresses uAUG-1 recognition equally well for poor and optimal context. By contrast, *E144R* and *R225K* impair recognition of the *SUI1* AUG only when it resides in poor context, suggesting that the effects of these Rps5 mutations are limited to the inherently weaker initiation site of native *SUI1* mRNA. This conclusion would be consistent with the fact that *E144R* and *R225K* also suppress recognition of the UUG start codon of *his4-301* mRNA, which resides in moderately strong context, as near-cognate codons are inherently weaker initiation sites even when present in optimum context.

Extending this last interpretation of *E144R* and *R225K*—that they preferentially discriminate against poor initiation sites—to explain our findings on leaky scanning of *GCN4* uAUG-1 would require a stipulation that uAUG-1 is a relatively weak initiation site even with the native optimum context present, which seems at odds with the fact that nearly all scanning ribosomes recognize native uAUG-1 in WT *GCN4* mRNA (*Grant et al., 1994*). However, one distinctive feature of WT uORF1 and

the elongated el.uORF1 contained in our leaky scanning reporter is the absence of typical coding sequences 3′ of the uAUG-1 start codon. WT uORF1 is only 3 codons long, and much of el.uORF1 is derived from non-translated triplets that normally reside between uORF4 and the *GCN4* coding sequence. This region has a relatively low propensity for secondary structure (*Kertesz et al., 2010*), which might enable elongating ribosomes to clear the initiation region more rapidly than occurs with more typical coding sequences. In fact, the properties of the leaky scanning reporter insure that the increases in *GCN4-lacZ* expression observed in the Rps5 mutants reflect diminished recognition of uAUG-1 relative to the main ORF start codon by scanning PICs. Considering that the *rps5* mutations appear to have no effect on recognition of the main ORF AUG in the *GCN4-lacZ* construct lacking el. uORF1 (*Figure 5*, row 7), the presence of greater secondary structure in the main ORF could reduce the rate of scanning through the initiation region to promote start codon recognition in a way that would be lacking at the el.uORF1 start codon. If elongating ribosomes located in the initiation region can enhance initiation by this mechanism, this could compensate for the reduction in $P_{IN}$ stability conferred by *E144R* and *R225K* for the *SUI1* and *GCN4* initiation regions in a way that would not occur for el.uORF1, making the latter sensitive to the destabilizing effects of these mutations even with optimum context present. Consistent with this scenario, it was shown recently that the presence of slowly-translated codons near the AUG codon can affect the initiation rate in yeast cells (*Chu et al., 2014*). Thus, it seems plausible that *E144R* and *R225K* decrease the efficiency of start codon recognition only for weak initiation sites, including near-cognate UUG codons and sub-optimal AUG start sites, without reducing recognition of AUG codons in strong context that initiate structured coding sequences.

The impaired recognition of the native *SUI1* AUG codon and attendant reduced synthesis of eIF1 conferred by the *rps5-E144R* and *-R225K* mutations evokes increased recognition of near-cognate, UUG start codons. While this elevated UUG:AUG initiation ratio is the expected outcome of diminished eIF1 abundance (*Ivanov et al., 2010*; *Martin-Marcos et al., 2011*), previously described Ssu⁻ substitutions in eIF1 and eIF1A reduce eIF1 levels by the same mechanism described here—discriminating against the weak context of the native *SUI1* AUG—but they suppress, rather than elevate, the UUG:AUG initiation ratio despite reduced eIF1 levels (*Martin-Marcos et al., 2011*). Thus, these eIF1 and eIF1A Ssu⁻ substitutions appear to have a stronger effect than *rps5-E144R* and *-R225K* in blocking selection of UUG start codons. On the other hand, the eIF1A and eIF1 Ssu⁻ substitutions confer much smaller increases in leaky scanning of *GCN4* uAUG-1 (*Fekete et al., 2007*) (Pilar Martin Marcos and AGH, unpublished observations) compared to that seen here for *rps5-E144R*, thus indicating a relatively greater defect in AUG recognition for the Rps5 mutation.

In addition to the mutations affecting the upper, structured portion of the β-hairpin loop (*E144R* and *R225K*), we also identified Ssu⁻ substitutions in the loop region that discriminate against UUG codons in the presence of Sui⁻ substitutions in eIF5 or eIF2β. Consistent with this, an exemplar of such mutations, *rps5-R148E*, specifically destabilized the $P_{IN}$ state formed at UUG, but not AUG, start codons in reconstituted PICs in vitro. The Rps5 loop substitutions do not discriminate against the weak context of the *SUI1* AUG codon, nor increase leaky scanning of el.uORF1 even when uAUG-1 resides in poor context, and thus exclusively destabilize PICs lacking a cognate (AUG) start codon. They differ from previously described Ssu⁻ substitutions in eIF1 (*Martin-Marcos et al., 2011*) in that the Rps5 substitutions efficiently suppress UUG initiation but do not discriminate against the poor context of the native *SUI1* AUG codon. This distinction might be explained by noting that eIF1 is the principal 'gate-keeper' that blocks utilization of weak initiation sites (*Hinnebusch, 2014*). As the eIF1 Ssu⁻ substitutions delay eIF1 release from the 40S subunit on start codon recognition (*Martin-Marcos et al., 2013*), they might discriminate more broadly against unstable PICs regardless of whether they lack strong context or a perfect codon:anticodon duplex in the P site. Our Rps5 loop substitutions, by contrast, appear to have a more nuanced effect that destabilizes $P_{IN}$ only when a mismatch occurs in the codon:anticodon duplex itself. As summarized in *Figure 9*, our results indicate that both E144 and R148 promote start codon selection by stabilizing the $P_{IN}$ state, and the finding that *E144R* reduces initiation at both UUG and sub-optimal AUG codons, while R148E impairs only UUG recognition, can be explained as the result of a relatively stronger contribution of E144 vs R148 to the stability of the $P_{IN}$ state.

There are several possibilities to explain how perturbing the Rps5 β-hairpin destabilizes the $P_{IN}$ state and reduces start codon recognition. The proximity of the hairpin loop to the E site (*Figure 2*) suggests a disruption of Rps5 contacts with the context nucleotides in mRNA. Indeed, R156 in the

loop interacts with the backbone of rRNA helix 23, which in turn contacts the −3 context nucleotide (*Figure 9—figure supplement 1*) (*Hussain et al., 2014*). If this interaction promotes the $P_{IN}$ state, it would help explain why loop residue substitutions impair recognition of UUG start codons (Ssu⁻ phenotype). However, except for the lethal substitution G151S, all of the substitutions affecting loop residues we examined—G151 through Q158—have weaker phenotypes compared to the E144R substitution in β-strand 1 of the hairpin itself, distant from the context nucleotides. Thus, perhaps structural alteration of the β-hairpin by E144R indirectly perturbs the conformation of the N-terminal tail of yeast Rps5, which promotes AUG recognition by its interaction with Rps16/uS9 (*Ghosh et al., 2014*), whose C-terminal tail closely approaches the codon-anticodon duplex in the P site (*Hussain et al., 2014*). Alternatively, E144R might affect the conformation or location of ribosomal proteins Rps28 and Rps14, also located in the exit channel and in contact with the Rps5 β-hairpin (*Hussain et al., 2014*), or of domain 1 of eIF2α, which interacts with other regions of Rps5 as well as Met-tRNA$_i$ in the $P_{IN}$ complex (*Figure 2*). In these latter scenarios, the inherent flexibility of the Rps5 hairpin loop could prevent loop substitutions from altering the orientation of the β-hairpin and attendant perturbations within the PIC compared to effects exerted by *E144R* or *R225K* on the structured portion of the hairpin.

The β-hairpin of uS7 also protrudes into the mRNA exit channel of bacterial ribosomes in position to interact with mRNA residues just upstream from the P site codon (*Jenner et al., 2007*). In bacterial elongation complexes, the hairpin is also in proximity to E-site tRNA, and truncation of the hairpin increases the frequency of frameshifting, most likely by allowing premature dissociation of the E-site tRNA (*Devaraj et al., 2009*). Interestingly, in the yeast py48S PIC, eIF2α-D1 essentially occupies the position of E-site tRNA (*Hussain et al., 2014*), in accordance with our suggestion that altering the β-hairpin of yeast uS7/Rps5 could impair start codon selection by perturbing the position or flexibility of eIF2α-D1. Regardless of the exact mechanisms involved, the strong impairment of AUG recognition in vivo and marked destabilization of the $P_{IN}$ state in vitro conferred by *E144R* dramatically illustrates that a 40S ribosomal protein functions as an equal partner with soluble initiation factors in ensuring efficient and accurate start codon recognition.

## Materials and methods

### Plasmids and yeast strains

Yeast strains and plasmids are listed in *Supplementary files 2, 3*, respectively.

Yeast strains used in this study are listed in *Supplementary file 2*. The $P_{GAL1}$-RPS5 strain JVY07 was generated from HLV01a (*MATa ura3-52 trp1Δ-63 leu2-3112 his4-301(ACG)*) by the one-step PCR strategy (*Longtine et al., 1998*) using the *kanMX4* cassette and selecting for resistance to kanamycin on rich medium containing galactose as carbon source (YPGal). Integration of the *kanMX:$P_{GAL1}$* promoter cassette at *RPS5* was verified by PCR analysis of genomic DNA using the appropriate primers. JVY07 was shown to be inviable on glucose medium (where the *GAL1* promoter is repressed) in a manner fully complemented by plasmid-borne *RPS5* alleles on pJV01 and pJV09. Derivatives of JVY07 harboring low copy *LEU2* plasmids containing WT (pJV09) or mutant *RPS5* alleles (pJV12-pJV53), listed in *Supplementary file 3*, were generated by transformation.

To avoid possible contamination with WT 40S subunits (from leaky expression of $P_{GAL1}$-RPS5 on glucose medium) transformants of JVY07 containing *rps5* alleles were not used for purifying mutant 40S subunits, and haploid strains harboring the relevant *rps5* alleles as the only source of Rps5 were generated for this purpose. Diploid strain F2009/YSC1021-672858 (*MATa/MATα ura3-Δ0/ura3-Δ0 leu2-Δ0/leu2-Δ0 his3Δ-1/his3Δ-1 lys2-Δ0/LYS2 met15-Δ0/MET15 rps5Δ::kanMX/RPS5*) was transformed with *URA3 RPS5* plasmid pJV38 and sporulated. Tetrads were dissected and analyzed for resistance to G418 to identify *rps5Δ::kanMX* ascospores, which was verified by PCR analysis of genomic DNA with appropriate primers. One such strain was selected as JVY11, and used as host to replace pJV38 with plasmids pJV09, pJV13 and pJV39, harboring *RPS5*, *rps5-E144R*, *rps5-R148E*, respectively, by plasmid-shuffling on medium containing 5-FOA (*Boeke et al., 1987*), resulting in strains JVY29, JVY15 and JVY52.

Plasmids used in this study are listed in *Supplementary file 3*. pJV01 was made by inserting into pRS315 a 1.6 kb BamHI restriction fragment containing *RPS5* flanked by 640 bp upstream and 320 bp downstream of the coding sequences that was amplified from genomic DNA of strain HLV01a. A BglII restriction site was introduced into pJV01 120 bp upstream of the *RPS5* ORF using the QuikChange

site-directed mutagenesis system (Agilent Technologies, Santa Clara, CA) to create pJV09, which was verified by DNA sequencing of the entire 1.6 kb insert. Introduction of the BglII site did not appreciably affect *RPS5* expression, as pJV09 and pJV01 were indistinguishable for complementation of strain JVY07 for growth on glucose medium. The insert from pJV09 was sub-cloned into pRS316 to create pJV38. *RPS5* fragments were amplified by fusion PCR to introduce the desired site-directed mutations, using primers listed in *Supplementary file 4* and pJV09 as template DNA. The mutagenized fragments were digested with BglII and NdeI and inserted between the same two restriction sites in pJV09, to produce pJV12-pJV52 (*Supplementary file 3*). Plasmid pJV53 was constructed similarly by using primers R225K, R225K_r and pJV13 (that was verified by sequencing) as template DNA. All constructs were verified by DNA sequencing of 1 kb from the inserted BglII site beyond the NdeI restriction site, covering the entire *RPS5* ORF.

## Biochemical analyses of yeast cells

Assays of β-galactosidase activity in whole-cell extracts (WCEs) were performed as described previously (*Moehle and Hinnebusch, 1991*). The sequence context of the start codon for both AUG and UUG *HIS4-lacZ* reporters is: 5′-AUA(AUG/UUG)G-3′. For Western analysis, WCEs were prepared by trichloroacetic acid extraction as described (*Reid and Schatz, 1982*), and immunoblot analysis was conducted as described previously (*Martin-Marcos et al., 2011*) with antibodies against eIF1 (*Valasek et al., 2004*) and Gcd6 (*Bushman et al., 1993*). Enhanced chemiluminescence (Amersham) was used to visualize immune complexes, and signal intensities were quantified by densitometry using NIH ImageJ software.

## Polysome profile analysis

For polysome analysis, strains were grown in SD + His + Ura + Trp at 30°C to $A_{600}$, ~1. Cycloheximide was added (50 µg/ml) 5 min prior to harvesting, and WCE was prepared in breaking buffer (20 mM Tris–HCl, pH 7.5, 50 mM KCl, 10 mM $MgCl_2$, 1 mM dithiothreitol, 5 mM NaF, 1 mM phenylmethylsulfonyl fluoride, 1 Complete EDTA-free Protease Inhibitor Tablet (Roche)/50 ml buffer). 15 $A_{260}$ units of WCE was separated by velocity sedimentation on a 4.5–45% sucrose gradient by centrifugation at 39,000 rpm for 3 hr in an SW41Ti rotor (Beckman). Gradient fractions were scanned at 254 nm to visualize ribosomal species.

## Biochemical analysis in the reconstituted yeast translation system

Initiation factors eIF1A and eIF1 were expressed in *Escherichia coli* and purified using the IMPACT system (NEB), and His6-tagged eIF2 was overexpressed in yeast and purified as described (*Acker et al., 2007*). WT and mutant 40S subunits were purified from yeast as described previously (*Acker et al., 2007*). Model mRNAs with the sequences 5′-GGAA[UC]7UAUG[CU]10C-3′ and 5′-GGAA[UC]7UUUG[CU]10C-3′ were purchased from Thermo Scientific. Yeast $tRNA_i^{Met}$ was synthesized from a hammerhead fusion template using T7 RNA polymerase and charged with [35S]-methionine or unlabeled methionine as previously described (*Acker et al., 2007*). $K_d$ values of TC (assembled with [35S]-Met-tRNAi) and 40S•eIF1•eIF1A•mRNA PICs, and rate constants of TC association/dissociation for the same PICs, were determined by gel shift assays as described previously (*Kolitz et al., 2009*) with the minor modifications described below.

### Buffers and reagents

For all experiments, the reaction buffer was 30 mM Hepes-KOH (pH 7.4), 100 mM potassium acetate (pH 7.4), 3 mM magnesium acetate, and 2 mM dithiothreitol. The composition of the enzyme buffer was 20 mM Hepes-KOH (pH 7.4), 100 mM KOAc (pH 7.4), 2 mM DTT, and 10% glycerol.

### Measurements of TC $K_d$ values in 40S·eIF1·eIF1A and 40S·eIF1·eIF1A·mRNA complexes

Gel shift assays were performed as described previously (*Kolitz et al., 2009*) with the following modifications. GDPNP·$Mg^{2+}$ was used at 100 µM, as this lower concentration was found not to reduce complex formation. TC was pre-formed for 15 min at 26°C before mixing with 40S subunits at various concentrations and the remaining factors. 10-fold concentrated stocks of 40S subunits were prepared by serial dilution. Final component concentrations in the reactions were: 1 nM [35S]-Met-tRNAi, 100 µM GDPNP, 200 nM eIF2, 1 µM each of eIF1 and eIF1A, and mRNA (when present) at 1 µM. Complexes

containing mRNA(AUG) were incubated at least 30 min at 26°C, whereas complexes lacking mRNA were incubated at least 45 min at 26°C. Total reaction volumes were 12 µl and were mixed with 3 µl of native gel dye (Acker et al., 2007) before resolving 13 µl by gel electrophoresis at 25 W for 30–45 min. Following electrophoresis, gel wells were washed to remove excess free [$^{35}$S]-Met-tRNA$_i$. The fraction of [$^{35}$S]-Met-tRNA$_i$ bound to 40S·eIF1·eIF1A or 40S·eIF1·eIF1A·mRNA complexes was measured using a PhosphorImager, plotted against the 40S subunit concentration, and the data were fit with a hyperbolic or quadratic binding equation, with the latter employed for tight binding.

## Kinetics of TC association and dissociation in 40S·eIF1·eIF1A·mRNA complexes

Measurements were carried out essentially as described previously (Kolitz et al., 2009). Reactions were performed in Recon buffer at final component concentrations of 250 nM eIF2, 1 nM [$^{35}$S]-Met-tRNA$_i$, 1 µM eIF1, 1 µM eIF1A, and 10 µM mRNA. Dissociation rates ($k_{off}$ values) were measured by monitoring the amount of labeled TC bound in 40S·eIF1·eIF1A·mRNA complexes over time using a native gel shift assay, as described above. 40S·eIF1·eIF1A·mRNA complexes were preassembled for 2 hr at 26°C in a reaction vol of 60 µl. Aliquots of 6 µl were removed at different times and mixed with 3 µl of a chase of unlabeled WT TC, containing 750 nM eIF2 and 300 nM Met-tRNA$_i$, representing a 300-fold excess over labeled TC. After addition of the chase to all time points, the reactions were mixed with native gel dye and loaded directly on a running native gel. A converging time course was employed so that all samples could be loaded simultaneously. The fraction of [$^{35}$S]-Met-tRNA$_i$ in 43S complexes was determined as described above and the data were fit with a single exponential equation. Association rates were measured by mixing labeled TC with 40S·eIF1·eIF1A·mRNA complexes and quenching the binding reaction at various times by adding a 300-fold excess of unlabeled WT TC. Reactions were assembled as described above using 6 µl of sample and 3 µl of chase, and completed reactions were mixed with 2 µl of native gel dye before resolving 10 µl by gel electrophoresis. As above, samples were loaded within min on a running native gel. The $k_{obs}$ values were calculated by plotting the fraction of [$^{35}$S]-Met-tRNA bound to 40S·eIF1·eIF1A·mRNA complexes against time and fitting the data with a single exponential equation. The resulting $k_{obs}$ values were plotted vs the 40S subunit concentrations used in different experiments and the data were fit to a straight line. The slopes of these lines correspond to the second-order rate constants ($k_{on}$) for TC binding.

## Acknowledgements

We thank Jagpreet Nanda for guidance with the in vitro reconstitution assays, and Jon Lorsch and members of our laboratories for helpful advice. This work was supported in part by the Intramural Program of the National Institutes of Health.

## Additional information

### Funding

| Funder | Grant reference | Author |
| --- | --- | --- |
| National Institutes of Health (NIH) | Intramural Program | Alan G Hinnebusch |

The funder had no role in study design, data collection and interpretation, or the decision to submit the work for publication.

### Author contributions

JV, Conception and design, Acquisition of data, Analysis and interpretation of data, Drafting or revising the article; YP, Conception and design, Acquisition of data; TED, AGH, Conception and design, Analysis and interpretation of data, Drafting or revising the article

## Additional files

### Supplementary files

• Supplementary file 1. Phenotypes of *RPS5* mutants.

- Supplementary file 2. Yeast strains employed in this study.

- Supplementary file 3. Plasmids employed in this study.

- Supplementary file 4. Oligonucleotide primers employed for mutagenesis in this study.

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
