## [Decision Letter]

Thank you for submitting your work entitled “The β-hairpin of 40S exit channel protein Rps5/uS7 promotes efficient and accurate translation initiation in vivo” for peer review at *eLife*. Your submission has been favorably evaluated by Randy Schekman (Senior editor), a Reviewing editor, and three reviewers.

The reviewers have discussed the reviews with one another and the Reviewing editor has drafted this decision to help you prepare a revised submission.

The following individuals responsible for the peer review of your submission have agreed to reveal their identity: Nahum Sonenberg (Reviewing editor); Matt Sachs and Sunnie Thompson (peer reviewers). A further reviewer remains anonymous.

Your manuscript represents an important contribution to the problem of accuracy mechanisms in protein synthesis, and the role of the small ribosomal subunit protein Rps5 in translation initiation site selection. By skillful extensive genetic and biochemical experiments, the authors identified a new element that assists recognition of the proper start codon during the initiation step in the eukaryotic model system of yeast. The data clearly indicate that the conserved β-hairpin and the C-terminal extension of ribosomal protein uS7 play a significant role in selection of the start codon while a pre-initiation complex scans messenger RNA to begin translation. The genuine finding that uS7 helps to recognize a correct initiation site in yeast is reminiscent of the well-known example of protein S12, which is important for accurate selection of tRNA during decoding in bacteria. The presented results center the attention on the idea that ribosomal proteins are not only ‘building blocks’ in the ribosome structure, but are also actively involved in its function together with numerous translation factors. The findings provide important and new insights into the molecular mechanism of how scanning eukaryotic ribosomes accomplish initiation site selection.

However, the reviewers raised several issues regarding the exact interpretation of some of the data, and the clarity of the arguments. You could consider adding additional transitional sentences, particularly when introducing more detailed aspects of initiation, and by breaking complex sentences into separate sentences. We hope that you will find the great majority of the comments to be useful in strengthening the paper. The comments of the three reviewers are listed below verbatim:

Reviewer #1:

Visweswaraiah et al. provide an important and well-executed study on the role of the small ribosomal subunit protein Rps5 in translation initiation site selection. The major findings are (i) specific mutations in the beta-hairpin region of this ribosomal protein, which from structural studies is near the site of start-codon recognition, have substantial impact on start codon selection and (ii) these mutations have a different spectrum of effects on start codon selection than do mutations of translation initiation factors. These analyses combine Saccharomyces genetics with biophysical approaches that use a reconstituted cell free system to address how the interplay between the ternary complex, the ribosome, and key initiation factors control start site selection. The findings provide important and new insights into the molecular mechanism of how scanning eukaryotic ribosomes accomplish initiation site selection.

The manuscript is well-written and organized but may be improved overall by considering where additional transitional sentences could help readers, particularly when introducing more detailed aspects of initiation, and by breaking complex sentences into separate sentences.

Specific comments are as follows:

Abstract: Include the idea of the relative importance of Rps5 relative to eIFs at the conclusion of the Abstract?

Introduction: The GTP bound to eIF2 […] the transition to this may be too abrupt in the first paragraph of the Introduction. Possibly give a more general overview into start site selection (as also occurs in subsequent paragraphs) and put this later in the Introduction?

Introduction, first paragraph: “AUG recognition triggers dissociation”. Use instead “start codon recognition triggers dissociation”.

Introduction, start of second paragraph: Define this TC-binding as loading so that the reference to loading is explicit and not inferred?

In the second paragraph of the subsection headed “Substitutions *E144R* and *R225K* impair translation initiation and start codon selection in vivo”: S223A mutation is also 2-fold in [Supplementary-material SD1-data].

In the same paragraph: I had trouble following the reasoning that the salt bridge might still be important although reinstating it had little impact on phenotype because the nature of the residues might be more important. Simply say that at this point that there is not direct evidence for the importance of the salt bridge?

Still in subsection “Substitutions *E144R* and *R225K* impair translation initiation and start codon selection in vivo”: Comment on *lacZ* reporter ratios or levels for these other mutants that enable UUG initiation from *his4-301* that confer Sui^-^ phenotypes? Then readers could get a sense of comparison for Figure 3 (also relevant for the second paragraph of the subsection headed “*E144R* and *R225K* elevate UUG initiation indirectly by exacerbating the effect of poor context of the *SUI1* start codon and thereby reducing eIF1 abundance”).

In the second paragraph of the subsection “*E144R* and *R225K* elevate UUG initiation indirectly by exacerbating the effect of poor context of the *SUI1* start codon and thereby reducing eIF1 abundance”: Is there a Slg^-^ phenotype for *R225K* in Figure 4—figure supplement 1? Also, in the lower panel, was the Y-axis of the image accidentally stretched (the colonies all look more oval than round).

In the last paragraph of the subsection headed “*E144R* and *R225K* confer leaky scanning of an upstream AUG start codon”: The authors appreciate that the effect on initiation of the elongated uORF1 UAUG1 relative to initiation at the *GCN4* AUG is complicated to interpret. Perhaps simply note that mechanistic possibilities are considered in the discussion and also move some of this material to the Discussion where these concerns are addressed?

In the last paragraph of the subsection headed “Ssu^-^ substitution *E144R* destabilizes the PIN conformation of the 48S PIC in vitro”: It is a minor point, but the connections between Figure 6—figure supplement 1 and Figure 4—figure supplement 2 due to possible specific effects of genotype changes is left open (specifically how *GCN4-lacZ* levels differ with the *RPS5* alleles with/without increased eIF1 dose is not clear).

In the first paragraph of the subsection headed “Substitutions in the loop region of the Rps5 β-hairpin increase fidelity of start codon 19 selection independently of context nucleotides”: *R157E* did not seem to have as substantial an effect (or I missed something).

Discussion, second paragraph: Impossible to increase or impossible to detect an increase?

Discussion, end of fourth paragraph: Are there possible ramifications for the tRNA anticodon's interactions with the start codon? This subject is discussed later.

Figure 1: Scanning inhibitor SI element in eIF1A NTT is not specifically indicated in the figure (it is in Figure 9).

Figure 2: State in legend that region of gradient where polysomes would have been is not shown (I assume they are not there)?

Figure 4—figure supplement 1: As mentioned above, *R225K* data look stretched for Y axis; if this is the case, then fix.

Reviewer #2:

The manuscript “The β-hairpin of 40S exit channel protein Rps5/uS7 promotes efficient and accurate translation initiation in vivo” represents an important contribution to the problem of accuracy mechanisms in protein synthesis. By skillful extensive genetic and biochemical experiments, the authors identified a new element that assists recognition of the proper start codon during the initiation step in the eukaryotic model system of yeast. The provided data clearly indicate that the conserved β-hairpin and the C-terminal extension of ribosomal protein uS7 play a significant role in selection of the start codon while a pre-initiation complex scans messenger RNA to begin translation. This information is undoubtedly valuable for ‘untangling’ complicated process of mRNA scanning during initiation in eukaryotes. The genuine finding that uS7 helps to recognize a correct initiation site in yeast is reminiscent of the well-known example of protein S12, which is important for accurate selection of tRNA during decoding in bacteria. The presented results center the attention on the idea that ribosomal proteins are not only ‘building blocks’ in the ribosome structure but are also actively involved in its function together with numerous translation factors. In my opinion, the manuscript fully deserves to be published by *eLife* because it solidly establishes a specific role of ribosome protein uS7 in initiation process.

My major concern is related to the results describing effects of mutations in uS7 on initiation from near-cognate UUG start codon. Because initiator Met-tRNAMet is expected to form less than three standard base-pairs with this codon, an initiation complex assembled on mRNA with UUG, a priori, is less stable than the complex formed on the AUG codon with complete three base-pairs. Consequently, the results describing, for example, dissociation of the initiation complexes assembled on the UUG codon can be exaggerated because inherently unstable complexes additionally fall apart owing to the experimental procedure (viz. electrophoresis during 45 min). Hence, Figure 6 shows that the initiation complex assembled on the UUG codon with the wild type 40S is very weak compared to the stable complex on the AUG codon. In this regard, it would be more correct to describe relative effects rather than provide exact values of constants. Since the Y-axis of most of the graphs is presented in arbitrary units, it would be also very helpful to show actual dmp/cpm values to demonstrate efficiency of signals and their differences.

I would also like to add the following suggestions:

1) It is not explained in the figures why eIF3 is not depicted in the schemes (Figures 1 and 9) and why in Figure 2 the authors provide a picture of the 48S initiation complex with only a few components while normal 48S contains much more initiation factors bound to the 40S subunit?

2) In Figure 2 I would recommend to use a more conventional orientation of 18S rRNA of the small ribosomal subunit, to clearly depict where exactly uS7 is located, where tRNAMet binds etc. I have the same problem with Figure 2: normally mRNA is shown with its 5'-nucleotides at the left and P-site codon on the right. In general, figure legends do not contain all necessary information.

3) The Discussion part is rather narrowed to a specific role of uS7 in eukaryotic initiation. It would be useful to draw some parallels with the role of the same protein in bacterial protein synthesis, whose conserved loop shares the same location relatively to the E- and P-site elements.

Reviewer #3:

The manuscript by Visweswaraiah et al., describes a functional role for *rps5* in start codon recognition. Previous studies have shown that *rps5* contacts the nucleotide at the -3 position (5' to the AUG start codon), which may suggest that rps5 affects start codon selection or fidelity. This manuscript examines the role of *rps5* on start codon recognition, both for recognition of the AUG and for the sequence context of the AUG. Their findings revealed that mutations in *rps5* resulted in a decrease in AUG recognition for start codons that were not in the optimal sequence context. As a result of this expression levels of eIF1, which uses a suboptimal start codon context to regulate its gene expression, are lower. When eIF1 levels were increased by expressing eIF1 from a plasmid, initiation at UUG returned to wild-type levels suggesting that the decrease in eIF1 is responsible for the increase in UUG initiation and not the *rps5* mutants. However, there must be some additional effect of *rps5* on UUG initiation since it was able to suppress UUG recognition in a yeast strain that harbored an eIF5 mutation that increased UUG initiation (Figure 4). The authors used binding studies to examine the affinity of the ternary complex bound to the Wild-type and *rps5* mutant 40S ribosomes. They show that the mutant *rps5* destabilizes the PIN/closed state. Interestingly, the transition from POUT to PIN is not affected, however, the stability of the PIN complexes is decreased for the *rps5* mutant ribosomes. Overall the findings are interesting and significant in that they suggest a novel role for *rps5* in start codon recognition, but the manuscript was a bit difficult to understand and it would benefit from additional clarifications and simplifications. Please see the following comments.

It is unclear why the *rps5* mutants promote leaking scanning for the *GCN4* reporter, since the phenotype is that these mutations confer a hyperaccurate initiation phenotype one would assume that a start codon in a good sequence context would be as good as WT initiation or better (the structural argument provided is not clear nor satisfying as written). However, it does appear that while the *rps5* mutants confer reduced initiation at start codons in a poor or week context that they may not enhance initiation in a favorable context. Could it be possible that a general decrease in frequency of start codon recognition could just have a much greater effect on the weaker or poor context AUGs and relatively less on the AUG in a good sequence context? Would this explain all the data, the binding studies showing reduced complex formation, the decrease in UUG initiation, and *SUI1* initiation? As the manuscript reads it appears that there are more than one model/mechanism for how *rps5* is affecting translation initiation depending on which reporter or assay is used. For example, when ribosomes with *rps5* mutations don't recognize the *GCN4* uAUG it is promoting leaky scanning, but when it doesn't recognize the *SUI1* or UUG it is hyperaccurate.

It is not clear how *rps5* could be affecting initiation at UUG codons given is position, perhaps the authors could discuss this further in the Discussion. Is the UUG in good sequence context or could this also be contributing to its initiation? What does moderately strong? Is the -3 position optimal? Perhaps defining these sequences would go a long way towards answering these questions.

There seems to be some disconnect between the Introduction and the Results where the authors state that *rps5* is required for efficient initiation. It is unclear where in the results the rate of initiation was measured. Perhaps the authors can better define what they mean by efficiency.

At the end of the second paragraph of the subsection headed “*E144R* and *R225K* elevate UUG initiation indirectly by exacerbating the effect of poor context of the *SUI1* start codon and thereby reducing eIF1 abundance” and Figure 4. The authors conclude: “the increased recognition of UUG start codons conferred by the *rps5* mutations is an indirect consequence of their reduced expression of eIF1.” They show that expressing eIF1 from a plasmid returns UUG/AUG recognition ratio back to WT and eIF1 levels increase (although quantification would be helpful as they appear not to come back to wt level). However, the authors assume but do not show whether the defect in sequence context is still present when eIF1 is expressed from a plasmid. Figure 4 should be repeated with yeast strains expressing eIF1 from a plasmid. <textboxend>

---

## [Author Response]

Reviewer #1:

[…] The manuscript is well-written and organized but may be improved overall by considering where additional transitional sentences could help readers, particularly when introducing more detailed aspects of initiation, and by breaking complex sentences into separate sentences.

*Specific comments are as follows*:

*Abstract*: *Include the idea of the relative importance of Rps5 relative to eIFs at the conclusion of the Abstract?*

We have included the relative importance of Rps5 in the Abstract and the sentence has now been changed to: “We conclude that the Rps5 β-hairpin is as crucial as soluble initiation factors for efficient and accurate start codon recognition.”

Introduction: The GTP bound to eIF2 […] the transition to this may be too abrupt in the first paragraph of the Introduction. Possibly give a more general overview into start site selection (as also occurs in subsequent paragraphs) and put this later in the Introduction?

To make the transition smoother we have changed the sentence as below:

“During scanning, the GTP bound to eIF2 in the TC is hydrolyzed in the 43S PIC in a manner dependent on the GTPase activating protein eIF5, but P_i_ release is blocked by eIF1, which also impedes stable binding of Met-tRNA_i_ in the P site.”

*Introduction, first paragraph: “AUG recognition triggers dissociation”. Use instead “start codon recognition triggers dissociation”*.

We have replaced AUG with start codon and the sentence now reads as “Start codon recognition triggers dissociation of eIF1”.

Introduction, start of second paragraph: Define this TC-binding as loading so that the reference to loading is explicit and not inferred?

We modified the sentence to: “…to which TC rapidly loads, bound in a state capable of inspecting successive triplets entering the P site…”

*In the second paragraph of the subsection headed “Substitutions* E144R *and* R225K *impair translation initiation and start codon selection* in vivo*”:* S223A *mutation is also 2-fold in*
[Supplementary-material SD1-data].

We wanted readers to note that *E144R* and 3 different substitutions of Arg255 are the only mutations that increased the UUG:AUG ratio substantially, which we defined as greater than two-fold, and the ratio for *S223A* does not exceed 2-fold.

*In the same paragraph: I had trouble following the reasoning that the salt bridge might still be important although reinstating it had little impact on phenotype because the nature of the residues might be more important*. *Simply say that at this point that there is not direct evidence for the importance of the salt bridge?*

We reasoned that combining the two mutations *E144R* and *R225E* both of which conferred >2-fold increases in the UUG:AUG ratio, should give an additive effect and increase the UUG:AUG ratio more than either of the single mutants. Since the double mutant did not increase the UUG:AUG ratio more than *E144R* alone, we inferred that reinstating the salt bridge mitigates to some degree the effects of the *E144R* single mutant. However, in deference to the referee, we added the sentence: “Additional experiments are needed to establish the importance of the salt bridge for Rps5 function” (Results).

*Still in the subsection “Substitutions* E144R *and* R225K *impair translation initiation and start codon selection* in vivo*”: Comment on* lacZ *reporter ratios or levels for these other mutants that enable UUG initiation from* his4-301 *that confer Sui*^*-*^
*phenotypes? Then readers could get a sense of comparison for*
Figure 3
*(also relevant for the second paragraph of the subsection headed “*E144R *and* R225K *elevate UUG initiation indirectly by exacerbating the effect of poor context of the* SUI1 *start codon and thereby reducing eIF1 abundance”)*.

We added the sentence: “For example, the eIF1 mutations *sui1-K37A* and *sui1-R33A* increase the UUG:AUG ratio by 4.8- and 7.7-fold, but only the latter suppresses the His^-^ phenotype of *his4-301*” (Results).

*In the second paragraph of the subsection headed “*E144R *and* R225K *elevate UUG initiation indirectly by exacerbating the effect of poor context of the* SUI1 *start codon and thereby reducing eIF1 abundance”: Is there a Slg*^*-*^
*phenotype for* R225K *in*
Figure 4—figure supplement 1*? Also, in the lower panel, was the Y-axis of the image accidentally stretched (the colonies all look more oval than round)*.

*R225K* has a Slg^-^ phenotype here when compared to the WT in row 1. To clarify the different extents of suppression by *scSUI1*, we added the sentence: “Introducing *scSUI1* also mitigated their Slg^-^ phenotypes, slightly for -*R225K* and substantially for -*E144R*”. The image has been adjusted to its original state in which the cell spottings and individual colonies appear round.

*In the last paragraph of the subsection headed “*E144R *and* R225K *confer leaky scanning of an upstream AUG start codon”: The authors appreciate that the effect on initiation of the elongated uORF1 UAUG1 relative to initiation at the* GCN4 *AUG is complicated to interpret. Perhaps simply note that mechanistic possibilities are considered in the discussion and also move some of this material to the Discussion where these concerns are addressed?*

We thank the reviewer for this suggestion and have removed this paragraph from the Results and integrated most of it into the Discussion.

*In the last paragraph of the subsection headed “Ssu*^*-*^
*substitution* E144R *destabilizes the PIN conformation of the 48S PIC* in vitro*”: It is a minor point, but the connections between*
Figure 6—figure supplement 1
*and*Figure 4—figure supplement 2
*due to possible specific effects of genotype changes is left open (specifically how* GCN4-lacZ *levels differ with the* RPS5 *alleles with/without increased eIF1 dose is not clear).*

This is a good point. We added the following sentence: “Derepression of *GCN4-lacZ* by *E144R* was evident even in the presence of *lcSUI1* (Figure 4—figure supplement 2, *E144R*/*scSUI1* vs. WT, unstarved), indicating that it does not result solely from the reduced eIF1 abundance in this mutant.”

*In the first paragraph of the subsection headed “Substitutions in the loop region of the Rps5 β-hairpin increase fidelity of start codon 19 selection independently of context nucleotides”:* R157E *did not seem to have as substantial an effect (or I missed something)*.

We agree that *R157E* did not significantly reduce the UUG:AUG ratio and have removed it from the text. The sentence now reads “Furthermore, *R148A/E*, *R156A/E* and *R157A* suppressed the elevated UUG:AUG ratio”.

Discussion, second paragraph: Impossible to increase or impossible to detect an increase?

We have changed the phrase to “it is impossible to detect an increase in leaky scanning”.

*Discussion, end of fourth paragraph: Are there possible ramifications for the tRNA anticodon's interactions with the start codon? This subject is discussed later*.

As the reviewer notes, the ramifications for the tRNA anticodon’s interaction with the start codon are discussed later in the Discussion.

Figure 1*: Scanning inhibitor SI element in eIF1A NTT is not specifically indicated in the figure (it is in*
Figure 9*)*.

Figure 1 now has the SI element in eIF1A indicated.

Figure 2: *State in legend that region of gradient where polysomes would have been is not shown (I assume they are not there)?*

The experimental conditions used in Figure 3 dissociate polysomes and 80S monosomes and allow separation of the ribosomal subunits. The legend now reads: “Similar to (C) but the cultures were not treated with cycloheximide and lysed in buffers without MgCl_2_ to allow separation of the dissociated ribosomal subunits.”

Figure 4—figure supplement 1*: As mentioned above,* R225K *data look stretched for Y axis; if this is the case, then fix*.

Addressed above.

Reviewer #2:

*[…] My major concern is related to the results describing effects of mutations in uS7 on initiation from near-cognate UUG start codon. Because initiator Met-tRNAMet is expected to form less than three standard base-pairs with this codon, an initiation complex assembled on mRNA with UUG,* a priori*, is less stable than the complex formed on the AUG codon with complete three base-pairs. Consequently, the results describing, for example, dissociation of the initiation complexes assembled on the UUG codon can be exaggerated because inherently unstable complexes additionally fall apart owing to the experimental procedure (viz. electrophoresis during 45 min). Hence,*
Figure 6
*shows that the initiation complex assembled on the UUG codon with the wild type 40S is very weak compared to the stable complex on the AUG codon. In this regard, it would be more correct to describe relative effects rather than provide exact values of constants. Since the Y-axis of most of the graphs is presented in arbitrary units, it would be also very helpful to show actual dmp/cpm values to demonstrate efficiency of signals and their differences*.

The in vitro reconstituted experiments in this study are based on the thorough analysis from the Lorsch lab described in [27], using single and double substitutions in the mRNA start codon to investigate the effects of base-pair mismatches with initiator tRNA on the rates of TC association and dissociation from 43S·mRNA complexes. They found that when UUG replaces AUG in the mRNA, the rate constant for TC binding (k_on_) was reduced by ∼40%, while the dissociation rate constant (k_off_) was increased ∼4-fold, to yield an increase in K_d_ of ∼7-fold. Thus, the referee is correct in stating that the UUG complex is inherently less stable than the AUG complex; however, this difference is the key feature of this assay that allows us to discern relatively greater effects of Ssu^-^ mutations on UUG versus AUG complexes. Based on the measured K_d_ values in [27], UUG is the near-cognate closest to AUG in forming a stable PIC. With WT ribosomes, the UUG complexes are still quite stable, with <20% dissociating in the first hour of incubation; and even less dissociation occurs when the ß-S264Y variant of eIF2 is employed, as done in Figure 8 where the destabilizing effect of *R148E* on UUG complexes was established. Note also that the key conclusion that *E144R* destabilizes the PIC derives in large part from its effect of increasing the k_off_ for the highly stable AUG complex (Figure 6). Dissociation of the UUG complexes during electrophoresis is not an issue because this would occur equally for all of the time points in the assay, including the zero-time point from which dissociation during the incubation is measured as a function of time. This assay has been used in several previous papers that we published in collaboration with the Lorsch lab, including the analysis of Sui^-^ and Ssu^-^ mutations in initiator tRNA (10), and of eIF1 Ssu^-^ mutations by [33]. Hence, we believe that the comparable analyses of *rps5* Ssu^-^ mutations presented here is both appropriate and accurate.

*I would also like to add the following suggestions*:

*1) It is not explained in the figures why eIF3 is not depicted in the schemes (*Figures 1 and 9*) and why in*
Figure 2
*the authors provide a picture of the 48S initiation complex with only a few components while normal 48S contains much more initiation factors bound to the 40S subunit?*

We did not include eIF3 in the schematic models of Figures 1 and 9 for several reasons. First, eIF3 is a large, multisubunit complex and accurately displaying even a simplified PIC containing only eIFs 1, 1A, TC and mRNA is already challenging. Second, the binding sites in the PIC for all of eIF3’s subunits are not yet known, and adding the well-established binding site for the PCI domains of eIF3α and -c on the solvent-exposed side of the 40S subunit would add little to visualization of the subunit interface surface depicted in these schematics. Third, the functions of the eIF3 subunits in scanning and AUG selection are not well established and Figures 1 and 9 are used to facilitate descriptions of the known functions of eIFs. Figure 2 (now Figure 2) is the partial yeast 48S structure from [22] where densities visible at high resolution were obtained only for the 40S, eIF1, eIF1A, mRNA, tRNA_i_ and eIF2α; nevertheless, it still represents the most complete PIC at reasonably high resolution currently available. We have altered Figure 2 to show the gamma subunit of eIF2 to indicate better the orientation of TC in the structure. The Figure 2 legend was modified to include: “Depiction of the partial yeast 48S PIC (PDB 3J81) showing Rps5 (gold), mRNA (orange), Met-tRNA_i_ (green), eIF2α (purple), eIF2γ (yellow), eIF1 (cyan) and eIF1A (blue). For clarity other ribosomal proteins, eIF2β and putative eIF5 densities are not shown.”

*2) In*
Figure 2
*I would recommend to use a more conventional orientation of 18S rRNA of the small ribosomal subunit, to clearly depict where exactly uS7 is located, where tRNAMet binds etc. I have the same problem with*
Figure 2*: normally mRNA is shown with its 5'-nucleotides at the left and P-site codon on the right. In general, figure legends do not contain all necessary information*.

We appreciate the reviewer’s point. While it might be non-standard, we chose the view in Figure 2 because it illustrates most clearly the location of uS7 in the mRNA exit channel, the residues identified in this study that affect initiation, and the position of the uS7 hairpin relative to the mRNA nucleotides and eIF2α-D1, which are all depicted similarly in Figure 2 after zooming in on the exit channel. We note that Figure 1 of Devaraj et al., on the effects of S7 hairpin mutations on frameshifting in *E. coli*, depicts exactly the same view of the bacterial components of the exit channel.

*3) The Discussion part is rather narrowed to a specific role of uS7 in eukaryotic initiation. It would be useful to draw some parallels with the role of the same protein in bacterial protein synthesis, whose conserved loop shares the same location relatively to the E- and P-site elements*.

We agree with the reviewer and have included the following sentences in the final paragraph: “The β-hairpin of uS7 also protrudes into the mRNA exit channel of bacterial ribosomes in position to interact with mRNA residues just upstream from the P site codon (24). In bacterial elongation complexes, the hairpin is also in proximity to E-site tRNA, and truncation of the hairpin increases the frequency of frameshifting most likely by allowing premature dissociation of the E-site tRNA (9). Interestingly, in the yeast py48S PIC, eIF2α-D1 occupies the position of E-site tRNA (22), in accordance with our suggestion that altering the β-hairpin of yeast uS7/Rps5 impairs start codon selection by altering the position or flexibility of eIF2α-D1.”

Reviewer #3:

*[…] The authors used binding studies to examine the affinity of the ternary complex bound to the Wild-type and* rps5 *mutant 40S ribosomes. They show that the mutant* rps5 *destabilizes the PIN/closed state. Interestingly, the transition from POUT to PIN is not affected, however, the stability of the PIN complexes is decreased for the* rps5 *mutant ribosomes. Overall the findings are interesting and significant in that they suggest a novel role for* rps5 *in start codon recognition, but the manuscript was a bit difficult to understand and it would benefit from additional clarifications and simplifications. Please see the following comments*.

*It is unclear why the* rps5 *mutants promote leaking scanning for the* GCN4 *reporter, since the phenotype is that these mutations confer a hyperaccurate initiation phenotype one would assume that a start codon in a good sequence context would be as good as WT initiation or better (the structural argument provided is not clear nor satisfying as written). However, it does appear that while the* rps5 *mutants confer reduced initiation at start codons in a poor or week context that they may not enhance initiation in a favorable context. Could it be possible that a general decrease in frequency of start codon recognition could just have a much greater effect on the weaker or poor context AUGs and relatively less on the AUG in a good sequence context? Would this explain all the data, the binding studies showing reduced complex formation, the decrease in UUG initiation, and* SUI1 *initiation? As the manuscript reads it appears that there are more than one model/mechanism for how* rps5 *is affecting translation initiation depending on which reporter or assay is used. For example, when ribosomes with* rps5 *mutations don't recognize the* GCN4 *uAUG it is promoting leaky scanning, but when it doesn't recognize the* SUI1 *or UUG it is hyperaccurate.*

We thank the referee for this suggestion. We agree with his/her interpretation and had actually laid the groundwork for advancing it as a likely scenario with the text in the second and third paragraphs of the Discussion that seeks to explain the increased leaky scanning observed for uAUG-1 in optimum context as the result of el.uORF1 being unstructured. However, at the end, we failed to propose this explanation explicitly. To rectify this, we added the following sentence: “Thus, it seems plausible that *E144R* and *R225K* decrease the efficiency of start codon recognition only for weak initiation sites, including near-cognate UUG codons and sub-optimal AUG start sites, without reducing recognition of AUG codons in strong context that initiate structured coding sequences.” And later in the Discussion we state: “As summarized in Figure 9, our results indicate that both E144 and R148 promote start codon selection by stabilizing the P_IN_ state, and the finding that *E144R* reduces initiation at both UUG and sub-optimal AUG codons, while *R148E* impairs only UUG recognition, can be explained as the result of a relatively stronger contribution of E144 versus R148 to the stability of the P_IN_ state”.

*It is not clear how* rps5 *could be affecting initiation at UUG codons given is position, perhaps the authors could discuss this further in the Discussion*.

We had presented various possibilities at the end of the Discussion of how mutations in the β-hairpin might affect start codon recognition. Briefly, we suggested a disruption of the interaction of hairpin loop residues with the rRNA residue in close proximity to the -3 context nucleotide, perturbation of the hairpin interaction with Rps16 whose CTT closely approaches the codon-anticodon duplex in the P-site, and altering the interaction with domain-1 of eIF2α that interacts with both tRNA_i_ and the mRNA 5’ of the start codon. We now suggest that this last mechanism would be analogous to the role of the orthologous hairpin in bacterial S7 in stabilizing the E-site tRNA during elongation, as eIF2α-D1 and E-site tRNA occupy very similar positions.

*Is the UUG in good sequence context or could this also be contributing to its initiation? What does moderately strong? Is the -3 position optimal? Perhaps defining these sequences would go a long way towards answering these questions*.

The sequence context for the *HIS4-lacZ* UUG reporter examined in vivo is strong, while that of the model mRNA used for in vitro experiments is weak. However, no effect of context has been observed thus far in the reconstituted system using the model mRNAs. Given the concordance of our findings in vivo with the *HIS4-lacZ* UUG reporter and those in vitro with the UUG model mRNA, we believe that context plays a minor role in suppression of UUG initiation by the Rps5 substitutions. Moreover, the context for the UUG and AUG *HIS4-lacZ* reporters are identical, and the same holds for the UUG and AUG model mRNAs, which is now stated explicitly in Results. The sequences of the model mRNAs used in the assays were mentioned in the Materials and methods, and we have now added the context of the *HIS4-lacZ* reporters to that section as well.

*There seems to be some disconnect between the Introduction and the Results where the authors state that* rps5 *is required for efficient initiation. It is unclear where in the results the rate of initiation was measured. Perhaps the authors can better define what they mean by efficiency.*

In Figure 3, we show that the *R225K* and *E144R* mutants exhibit moderate, but statistically significant reductions in the polysome:monosome ratio, which indicates a reduction in the rate of initiation versus elongation. We have cited this result in the first paragraph of the Discussion as evidence that the *E144R* substitution in β-strand 1 of the hairpin reduces the rate of bulk translation initiation. Combining this result with the increased leaky scanning of *GCN4* uAUG-1 conferred by this mutation leads us to conclude that selection of AUG codons by the scanning PIC is less efficient in this Rps5 mutant.

*At the end of the second paragraph of the subsection headed “*E144R *and* R225K *elevate UUG initiation indirectly by exacerbating the effect of poor context of the* SUI1 *start codon and thereby reducing eIF1 abundance” and*
Figure 4*. The authors conclude: “the increased recognition of UUG start codons conferred by the* rps5 *mutations is an indirect consequence of their reduced expression of eIF1.” They show that expressing eIF1 from a plasmid returns UUG/AUG recognition ratio back to WT and eIF1 levels increase (although quantification would be helpful as they appear not to come back to wt level). However, the authors assume but do not show whether the defect in sequence context is still present when eIF1 is expressed from a plasmid.*
Figure 4
*should be repeated with yeast strains expressing eIF1 from a plasmid*.

We have quantified the blots and added the results to Figure 4, and the reviewer is correct that the levels of eIF1 are not fully restored to the WT level when an extra copy of *SUI1* under its native promoter is introduced into the mutants on a single-copy plasmid (compare lanes 3-4 and 7-8 to 9-10). Despite this, the increase in eIF1 levels in the mutants achieved with *scSUI1* is sufficient to restore the elevated UUG/AUG ratio to the WT level (Figure 4). This demonstrates that elevated UUG initiation is an indirect consequence of low eIF1 levels in the mutants, which is the main point of this experiment. The fact that the eIF1 levels in the mutants harboring *scSUI1* remain well below those seen in the corresponding WT/*scSUI1* transformant (lanes 3-4 & 7-8 vs. 11-12) indicates that enhanced discrimination against the poor context of native *SUI1* still occurs in the mutants at the boosted levels of eIF1. Similarly, we show later in the paper that the increased leaky scanning of *GCN4* uAUG-1 in the mutants is not diminished by increasing eIF1 levels with *scSUI1*. Thus, we think it is clear that the defects in AUG recognition, whether for the native *SUI1* AUG or *GCN4* uAUG-1, do not arise from the reduced eIF1 levels in the mutants. In fact, because eIF1 promotes scanning, reduced eIF1 levels per se should diminish discrimination against the poor context of SU1I and boost eIF1 levels; and if anything, should reduce, not increase, leaky scanning of uAUG-1.